# Enzyme adaptation to habitat thermal legacy shapes the thermal plasticity of marine microbiomes

Ramona Marasco [1,11], Marco Fusi [1,2,11], Cristina Coscolín [3,11], Alan Barozzi [1], David Almendral[3], Rafael Bargiela[4], Christina Gohlke neé Nutschel[5], Christopher Pfleger[6], Jonas Dittrich [6], Holger Gohlke[5,6,7], Ruth Matesanz[8], Sergio Sanchez-Carrillo[3,9], Francesca Mapelli [10], Tatyana N. Chernikova[4], Peter N. Golyshin [4], Manuel Ferrer [3] ✉ & Daniele Daffonchio [1] ✉

Microbial communities respond to temperature with physiological adaptation and compositional turnover. Whether thermal selection of enzymes explains marine microbiome plasticity in response to temperature remains unresolved. By quantifying the thermal behaviour of seven functionally-independent enzyme classes (esterase, extradiol dioxygenase, phosphatase, beta-galactosidase, nuclease, transaminase, and aldo-keto reductase) in native proteomes of marine sediment microbiomes from the Irish Sea to the southern Red Sea, we record a significant effect of the mean annual temperature (MAT) on enzyme response in all cases. Activity and stability profiles of 228 esterases and 5 extradiol dioxygenases from sediment and seawater across 70 locations worldwide validate this thermal pattern. Modelling the esterase phase transition temperature as a measure of structural flexibility confirms the observed relationship with MAT. Furthermore, when considering temperature variability in sites with non-significantly different MATs, the broadest range of enzyme thermal behaviour and the highest growth plasticity of the enriched heterotrophic bacteria occur in samples with the widest annual thermal variability. These results indicate that temperature-driven enzyme selection shapes microbiome thermal plasticity and that thermal variability finely tunes such processes and should be considered alongside MAT in forecasting microbial community thermal response.

As the Earth's climate changes at a rapid pace, forecasting ecological patterns has become increasingly important[1]. Temperature is an environmental and climatic variable[2,3] that can severely impact sensitive ecosystems because it is a strong selective force that acts on the biology of a cell[4-6]. Temperature selects and shapes the cellular building blocks, including enzymes, that maintain vital physiological and cellular processes[7,8].

At the cellular level, the plasticity of thermal response originates from different strategies of adaptation[9-12]: (i) physiological plasticity (acclimation), defined as the extent to which an organism can change its physiology in response to environmental cues (e.g., increase in enzyme concentration in response to thermal change); (ii) regulation of genes (e.g., temperature-dependent expression of isoenzymes and/or epigenetic regulation); and (iii) genetic

adaptation that drives the selection of new enzyme variants for which reaction rate is adapted to changing environmental conditions (e.g., advantageous mutations or acquisition of new genes). The latter mechanism is particularly important in organisms with a short generation time (and high turnover), such as microorganisms, that are capable of timely adaptation to new conditions[13]. The combination of these strategies defines the final organismal metabolic and physiological performance[7,8,14,15].

At the community level, the above strategies are implemented by ecological sorting of species, species turnover and environmental filtering[13,16,17], implying the rearrangement of community composition through, for example, the replacement of dominant species with adapted types with improved fitness. The coalescence of such strategies at cellular and community levels determines the plasticity of the environmental communities, their diversity, stability, and functionality[18-20]. For instance, the diversity and distribution of up to 60% of the global ocean microbiome are associated with temperature[21].

The mean annual temperature (MAT) of a given location is the parameter most commonly used to test the response of communities, specifically their components and molecular building blocks, to temperature[22,23]. However, organisms and their components experience temperature fluctuation over time[24-27] rather than MAT, which is defined as the temperature variability that best describes the thermal legacy to which they are exposed. In complex microbial communities, the selection of proteins and enzymes in response to the thermal legacy should occur across the whole taxonomic range to support growth and functional plasticity[28]. However, whether enzyme thermal properties vary systematically in response to changes in temperature and how the thermal legacy (temperature variability or temperature fluctuation) affects the plasticity of microbial communities remains unresolved.

Here, we hypothesize that environmental MAT contributes to drive the adaptability of marine microbial communities by tuning the thermal plasticity of their enzymes, and that such plasticity is fine-tuned further by environmental temperature variability.

To test this hypothesis, we studied the enzymatic thermal response of marine microbial communities at a global geographical scale across different locations with different thermal conditions (Supplementary Fig. S1). We first determined the optimal temperature for maximum activity ($T_{opt}$) of seven functionally-independent enzyme classes (esterase, phosphatase, beta-galactosidase, nuclease, transaminase, extradiol dioxygenase (EXDO) and aldo-keto reductase) in the total active proteomes recovered from marine sediments across a broad latitudinal gradient, ranging from the Irish Sea (MAT, 12 °C) to the southern Red Sea (MAT, 30 °C). We further analysed the response of diverse purified enzymes retrieved from sediment and seawater metagenomes to MAT, ranging from −1.4 °C to 29.5 °C, from about 70 worldwide locations. A total of 228 esterases and 5 EXDOs were targeted to measure $T_{opt}$ and to determine the thermostability and structural rigidity properties. To assess the effect of temperature variability on the thermal plasticity of marine microbial communities, we determined the response of the esterases in the total active proteomes to temperature and the growth of three bacterial communities in sediments with the same MAT but with a variable thermal legacy, in terms of temperature variability, throughout the year.

Our results show that marine microbiome thermal plasticity is reflected in cellular enzyme selection driven by the environmental thermal legacy: the selection of thermally adapted enzymes in microbial communities is explained by MAT, but the breadth of thermal plasticity of the microbiome and their enzymes (tested as esterases) is shaped and fine-tuned by the temperature variability of the habitat.

## Results and discussion

### MAT is a major driver of marine microbiome enzymatic activity

To test whether enzyme adaptation to habitat temperature (i.e., physiological plasticity and/or genetic variation) explains the assembly of marine microbiomes, we quantitatively evaluated: (i) the temperature range at which microbial enzymes are active (thermal profile) and (ii) their $T_{opt}$ in sediments with different MAT. We extracted the total active proteins from the microbial communities inhabiting sediments collected from 14 locations across a latitudinal transect (Supplementary Table S1), ranging from the Irish Sea (MAT, 12 °C; latitude 53°N) to the southern Red Sea (MAT, 30 °C; latitude, 16°N), and evaluated the enzymatic activity of seven enzyme classes, namely esterases, phosphatases, transaminases, EXDOs, beta-galactosidases, nucleases and aldo-keto reductases (Fig. 1, Supplementary Data S1 and Source Data). A significant variation in the thermal profiles across the transect was found for all the enzymatic activities tested (range $R^2$, 0.51–0.80, all $p < 0.01$). Proteins from the warmer sites (Red Sea, MAT: 30 °C) showed the highest activity at higher temperatures, being most active (100% activity) at temperatures between 40 °C and 60 °C. Conversely, proteins from colder sediments (Irish Sea, MAT: 12 °C) were most active at lower temperatures, between 8 °C and 30 °C (average activity at 8 °C = 32.2% [min = 1.8% for beta-galactosidases, Fig. 1e; max = 100% for aldo-keto reductases, Fig. 1g]). Considering esterase activity (Fig. 1a), for example, the difference in $T_{opt}$ between the coldest and warmest sites was 35 °C (max 55 °C and min 20 °C in the Red Sea and Irish Sea sediments, respectively), while sediments collected from the Mediterranean Sea showed intermediate profiles (Fig. 1). Notably, the variation in the thermal profile for these enzymatic activities (i.e., $T_{opt}$) was mainly explained by MAT rather than other environmental variables, such as salinity and pH (esterases in Table 1; other enzymes in Supplementary Table S2).

To evaluate whether the efficiency of protein extraction could influence the correlation between MAT and $T_{opt}$, and thus the diversity of active enzymes, we performed a comprehensive analysis of the proteomes extracted from 7 out of 14 sediments by two-dimensional sodium dodecyl sulphate polyacrylamide gel electrophoresis (SDS-PAGE) and Liquid Chromatography-Electrospray Ionization Tandem Mass Spectrometric (LC-ESI-MS/MS) analysis combined with metagenome analysis. The results revealed that the recovered proteomes were not dominated by a specific group/type of proteins (Supplementary Fig. S2). For a wide range of proteins, the relative abundance—referred to as the total number of open reading frames (ORF) mapped in the relative metagenomes—was 1.5% (interquartile range (IQR): 1.09%) (Supplementary Table S3 and Supplementary Data S2), with no significant differences among samples and no correlation with MAT ($p = 0.24$). Having confirmed that there were no differences in the relative abundance of proteins identified in the proteomes, we then proceeded to estimate the diversity and relative abundance of genes responsible for the activity measured in the corresponding metagenomes. We focused this quality check control on esterases because of the availability of a curated database, the Lipase Engineering Database[29], which facilitates their identification through DIAMOND-BLASTP search tool[30]. We identified a total of 947 sequences potentially encoding esterases, of which none were present in all samples and only 73 shared among a few samples (Supplementary Fig. S3). These sequences had a relative abundance of 0.10% to 0.29% compared to the total ORFs (average: 0.22%, IQR: 0.08%; Supplementary Table S3 and Supplementary Data S2), with no correlation with site MAT ($p = 0.098$).

The OMICS analyses showed that the MAT-dependent activity response of the seven enzyme classes was not biased by the protein extraction efficiency and did not depend on the relative abundance of the enzymes responsible for the activity, but rather that each geographical location selects specific sets of enzymes with distinct thermal characteristics. This finding provides a physiological explanation

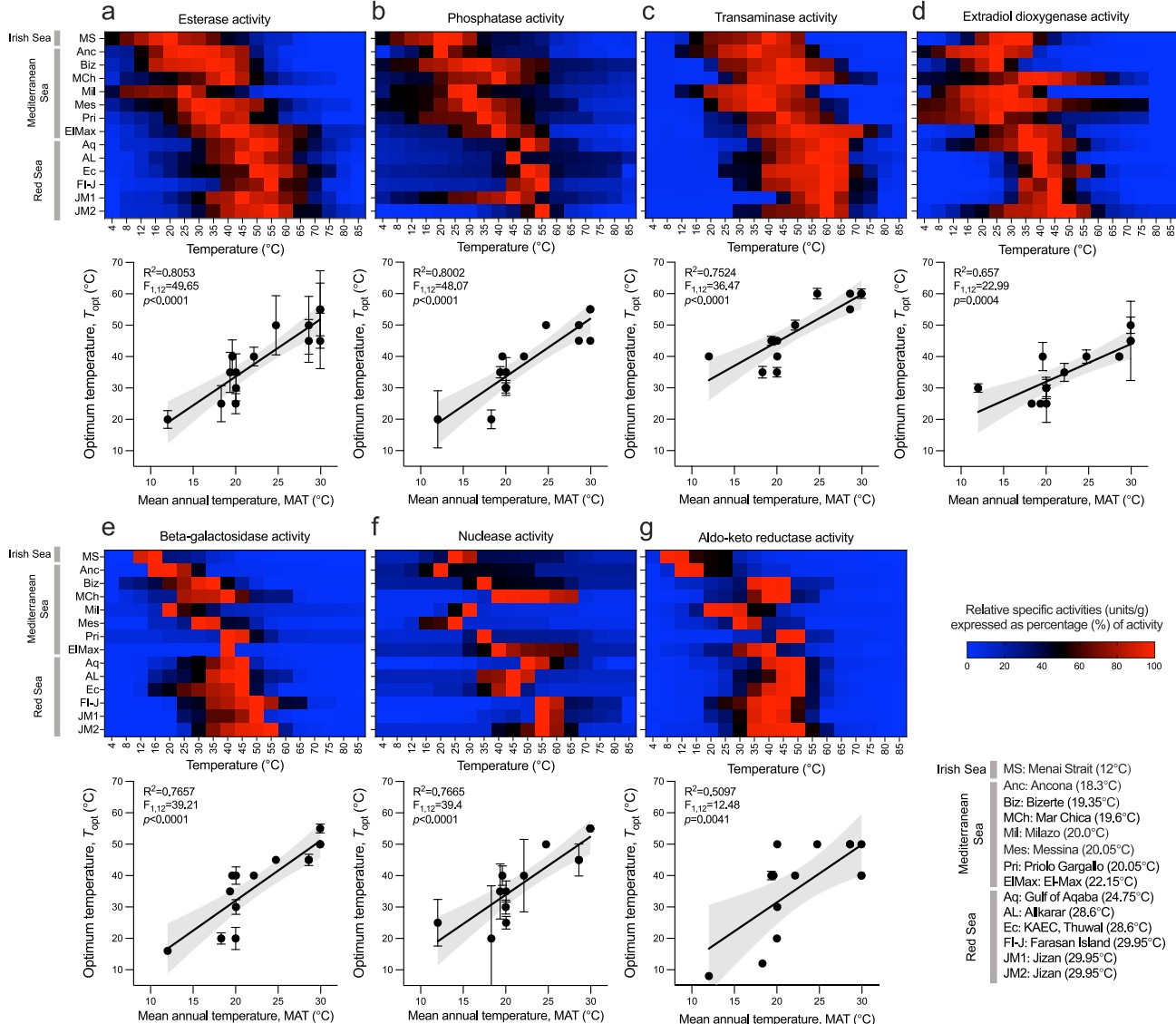

**Fig. 1 | Thermal adaptation of microbial community isozymes from sea sediments. a–g** Thermal profiles for **a** esterase, **b** phosphatase, **c** transaminase, **d** extradiol dioxygenase, **e** beta-galactosidase, **f** nuclease, and **g** aldo-keto reductase activities of total microbial proteins, in their active form, extracted from microbial communities inhabiting marine sediments from different locations encompassing the Irish Sea, Mediterranean Sea and Red Sea (Supplementary Table S1). Heatmaps represent the relative percentages (%) of specific activity at each temperature calculated from the initial enzymatic rate (units/mg; raw data in Supplementary Data S1) compared with the maximum activity (mean of three triplicates). The colour code ranges from intense blue (no activity, 0%) to red (100% activity). Under each heatmap, the relationship between site temperature (°C) and temperature of maximum enzymatic activities ($T_{opt}$) expressed as mean ± SD ($n = 3$) is reported for each enzymatic activity (plotted data are reported in Source Data). Simple linear regressions are plotted as black lines with grey zone representing the 95% confidence intervals.

for the evidence that temperature drives the diversity and distribution of microbiomes in the ocean[21]. Similar to findings at the organismal level[7,8], enzymes act as key cellular building blocks selected by the thermal regime at the metaorganismal level of the marine microbial communities. Since the same responses to MAT were observed for seven different functionally-independent classes of enzymes, we speculate that this may be a general feature of the proteomes of marine microbial communities. The broad enzymatic diversity identified by the proteomic and metagenomic surveys and the variability of the enzyme thermal responses across locations suggests that genetic adaptation (e.g., the occurrence of mutations and the formation of new proteins) coupled with thermal selection and species turnover (e.g., enrichment of variants/taxa that arise as soon as the temperature conditions allow) model the thermal plasticity and diversity[21] of marine microbiomes. These data indicate that the selection of adapted

enzyme variants contributes to drive the functional plasticity of the microbial communities; however, based on our data, we cannot define which of the adaptive mechanisms prevail.

## MAT explains the thermal response of cloned marine enzymes

To ascertain whether the temperature-dependent profiles of the seven enzyme classes analysed from total active proteomes were also valid for individual enzymes, we examined the $T_{opt}$ and denaturing temperature ($T_d$) of 78 esterases and 5 EXDOs retrieved by metagenomics screening (an average of eight enzymes per site; max: 17; min: 3; Supplementary Data S3). We chose esterases as they are widely distributed in nature within microbial communities (at least one is found in every bacterial genome), and multiple optimized assays are available[31,32]. The proteins (average pairwise sequence identity: 16.68%, IQR: 6.70%) were recovered from a subset of ten sediments along the

**Table 1 | Best linear models describing the effects of temperature (MAT), pH and salinity on the temperature for maximum activity ($T_{opt}$) of esterases in: (i) the total active proteins extracted from sediment along the transect from the Irish Sea to the southern Red Sea (see Fig. 1a); (ii) 78 individual esterases of sediment/seawater microbial communities from a subset of ten marine locations selected from the latitudinal transect in (i) (see Fig. 2a); and (iii) 150 individual enzymes from the seawater Tara Ocean dataset (56 locations; see Fig. 2c)**

| Samples | Model | Residual d.f. | $R^2$ | AIC |
|---|---|---|---|---|
| (i) Total active proteins from sediment transect $n = 14$ (see Fig. 1a) | Intercept | 1,13 | — | 110.8619 |
| | **Temperature** | **1,12** | **0.8128** | **89.0087** |
| | pH | 1,12 | 0.29 | 89.92584 |
| | Salinity | 1,12 | 0.043 | 91.00381 |
| | Temperature + Salinity | 2,11 | 0.8009 | 89.24695 |
| | Temperature + pH | 2,11 | 0.795 | 111.87853 |
| | Salinity + pH | 2,11 | 0.8104 | 111.81935 |
| | Temperature + Salinity + pH | 3,10 | 0.045 | 110.6288 |
| (ii) Individual esterases from sediment transect $n = 78$ (see Fig. 2a) | Intercept | 1,77 | — | 679.2983 |
| | **Temperature** | **1,76** | **0.5735** | **609.5436** |
| | pH | 1,76 | 0.2803 | 611.2151 |
| | Salinity | 1,76 | 0.3449 | 613.1814 |
| | Temperature + Salinity | 2,75 | 0.5699 | 611.4951 |
| | Temperature + pH | 2,75 | 0.5647 | 646.1708 |
| | Salinity + pH | 2,75 | 0.5685 | 652.9775 |
| | Temperature + Salinity + pH | 3,74 | 0.3447 | 645.1768 |
| (iii) Individual enzymes from seawater, TARA Ocean dataset $n = 150$ (see Fig. 2d) | Intercept | 1,149 | — | 679.2983 |
| | **Temperature** | **1,148** | **0.5735** | **609.5436** |
| | pH | 1,148 | 0.5647 | 611.2151 |
| | Salinity | 1,148 | 0.3449 | 613.1814 |
| | Temperature + Salinity | 2,147 | 0.5699 | 611.4951 |
| | Temperature + pH | 2,147 | 0.5685 | 646.1708 |
| | Salinity + pH | 2,147 | 0.3447 | 652.9775 |
| | Temperature + Salinity + pH | 3,146 | 0.4447 | 645.1768 |

The residual degrees of freedom (d.f.) are given. The treatment degrees of freedom and sum of squares only apply to the term that was added to the model. The Akaike information criterion (AIC) was calculated for each model and the lowest AIC (in bold) indicates the best model obtained.

latitudinal gradient from the Irish Sea (53°N) to the northern Red Sea (30°N). However, we could not obtain enzymes from the original Irish Sea samples (MAT, 12 °C); these were obtained from a 12-month enrichment culture incubated at 20 °C. Overall, the MATs examined in this experiment ranged from 18.3 °C to 24.8 °C. We found a significant positive relationship between the $T_{opt}$ of the cloned enzymes and the MAT of the locations from which the enzymes were derived ($R^2 = 0.57$, $p = 3.4E-12$; Fig. 2a), as well as between $T_d$ and MAT ($R^2 = 0.81$, $p = 6.9E-14$; Fig. 2b); no significant relationship was found between esterase thermal response ($T_{opt}$ and $T_d$) and salinity or pH (Table 1; Supplementary Fig. S4). This analysis was extended to five EXDOs, which are structurally and catalytically different from esterases. The positive relationships of $T_{opt}$ and $T_d$ with MAT were also confirmed for the EXDOs (Supplementary Fig. S5).

To enlarge the range of MATs explored in the analysis, we also included a set of 150 esterases retrieved from seawater metagenomes from 56 locations available in the Tara Ocean dataset, which have MATs ranging from −1.4 °C to 29.5 °C (latitudes 62.2°S to 43.7°N; Supplementary Data S4). The average pairwise sequence identity of esterases was 38.1% (IQR: 18.6%), and none of the sequences showed 100% identity. MAT was also the best predictor of the observed thermal response of Tara Ocean enzymes for both $T_{opt}$ and $T_d$ (Fig. 2c, d;

Supplementary Figs. S6 and S7), and neither the addition of salinity or pH did improve the power of the model (Table 1). However, the relationship of $T_{opt}$ and $T_d$ with MAT was not linear. This was highlighted by the significant piecewise regression, which indicates an increased response of the enzymes at the highest MATs. MAT breakpoints were recorded at 27.7 °C and 25.1 °C for $T_{opt}$ and $T_d$, respectively (Fig. 2c, d). Before the breakpoints, both $T_{opt}$ and $T_d$ slightly increased with increasing MAT (indicated by a flatter slope, $T_{opt}$: 0.45 and $T_d$: 0.54), while beyond the breakpoint, both $T_{opt}$ and $T_d$ sharply increased with MAT (indicated by a steeper slope of the line, $T_{opt}$: 6.48 and $T_d$: 4.45). This offers the advantage of an increased potential for species adaptation at higher temperatures.

The analysis of the phylogenetic clustering of the 228 esterases herein characterized did not show specific grouping based on MAT, but rather by enzyme families (Supplementary Fig. S8; Supplementary Note S1). The low pairwise sequence similarity observed among sequences (16.7% and 38.1% among sequences from the Irish Sea–Red Sea transect and the Tara Ocean metagenomes, respectively) indicated that the diversity of the investigated esterases was not dominated by a particular type or highly similar protein clusters, but instead comprised of different non-redundant sequences assigned to multiple folds, families, and subfamilies that, in many cases, are distantly related to known homologues.

Since changes in protein flexibility may play a role in the thermal adaptation to different temperatures without altering the global structure and the active site[33], we analysed this parameter in our esterases. Biomolecular thermostability can have a thermodynamic or kinetic origin[34]. Thus far, rigidity analysis has been used to investigate structural effects on the folded state only, and it has been estimated that increased structural rigidity is responsible for increased thermostability in 60% of cases[35]. Furthermore, rigidity analysis cannot account for the time-dependency of processes[36]. Constraint Network Analysis (CNA)-based analyses of the relationship between structural rigidity and flexibility versus thermostability have been applied on pairs[33,37] and series of homologous proteins[38,39] from psychrophilic to hyperthermophilic organisms, as well as on a series of variants from one protein retro-[40] and prospectively[35,41]. Here, we observed significant correlations between the computed phase transition temperature ($T_p$), a measure for global structural rigidity, and MAT (Irish Sea–Red Sea transect: $R^2 = 0.12$, $p = 0.0034$, Fig. 2e; Tara Ocean: significant regression only after MAT breakpoint of 21.6 °C, $R^2 = 0.08$, $p = 0.0096$, Fig. 2f). The absence of a significant regression in $T_p$ for esterases with environmental MAT around and below ~20 ± 1 °C could be linked to the fact that the structural ensembles for the rigidity analyses by molecular dynamics (MD) simulations were generated at room temperature, leading to an evening out of $T_p$ values for esterases with a MAT that transition around or below this temperature. However, we cannot exclude that the ~20 ± 1 °C trade-off represents the onset of evolutionary trade-offs that may occur during biochemical adaptation to lower temperatures, where enzymes have to keep a minimum rigidity for correct functioning[7]. Overall, these findings indicate that esterases from microorganisms found in environments with higher MAT have evolved so that these esterases are more rigid (less flexible). This might mirror the principle of corresponding states, according to which homologs from mesophilic and thermophilic organisms have similar flexibility and rigidity characteristics at their respective growth temperatures[33,37].

Although we did not examine enzyme activities at locations representing all marine thermal regimes, ranging from polar sites to hydrothermal vents, our results show that the response of marine microbial community enzymes to temperature is dictated by the environmental temperature (i.e., MAT). This finding supports the assumption that the adaptation of microbial communities is primarily driven by the thermal plasticity of microbial enzymatic machinery. This also explains why temperature is consistently found to be one of

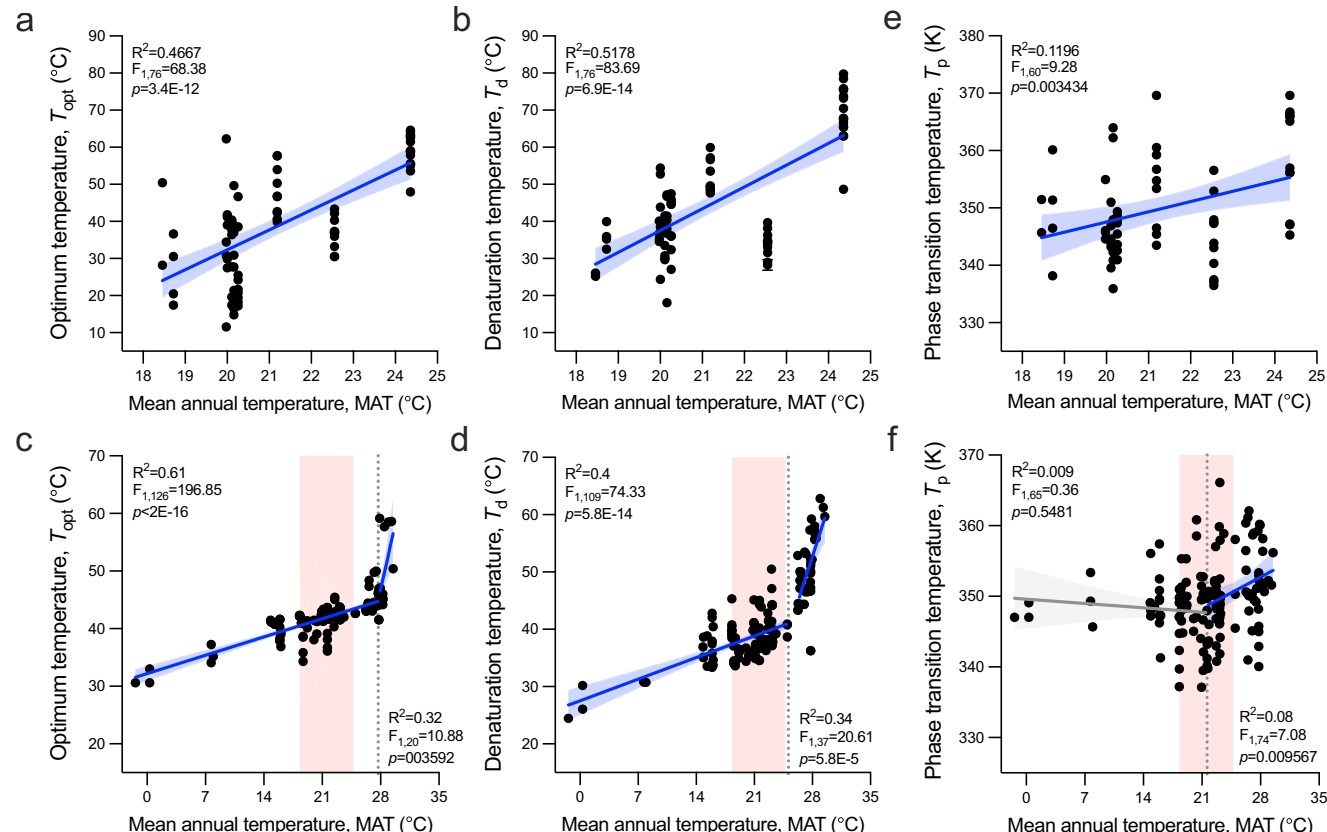

**Fig. 2 | Thermal adaptation of purified isozymes from seawater samples.**
**a** Optimal temperature ($T_{opt}$), **b** denaturation temperature ($T_d$), and **e** phase transition temperature ($T_p$) patterns as a function of the MAT at the site from which the 78 esterases originated along the North–South latitudinal transect. **c** $T_{opt}$, **d** $T_d$, and **f** $T_p$ patterns of 150 esterases from 56 TARA ocean locations (Supplementary Data S4) as a function of the MAT at the site. $T_{opt}$, $T_d$, and $T_p$ of esterases are determined by measuring the initial hydrolysis rate of 4-nitrophenyl-propionate, the CD ellipticity changes (in millidegrees, mdeg; θ) at 220 nm at different temperatures at a rate of 0.5 °C per min, and performing Constraint Network Analysis (CNA), respectively. $T_{opt}$ and $T_d$ values are plotted as mean ($n = 3$) and related SD

are reported in Supplementary Data S3 and S4. $T_p$ values are presented as the mean of five independent MD simulations analysed with CNA and related SEM are given in Supplementary Data S3 and S4. The linear regressions are performed using a two-sided test in R; $R^2$, degrees of freedom, F and $p$-values are reported in each graph. Significant regressions are reported as blue lines, while non-significant in grey. The blue/grey zone represents the confidence value of 95%. In the case of esterases from the TARA ocean dataset, piecewise regressions were run, and the breakpoints (flexus) where the slope of the regressions significantly changed are indicated with dashed lines on the MAT axis. Red boxes in panels **c**, **d** and **f** indicate the MAT range covered by the Irish Sea–Red Sea transect in panels **a**, **b** and **e**.

the main drivers shaping enzymatic activity and stability, microbial community diversity and metabolism[5,21–23,42] across broad geographical ranges. The relationships with MAT were confirmed by both total active proteins extracted from environmental samples (where expression could play a role[43]) and in individual enzymes from different sources. Therefore, we consider that the level of expression (physiological plasticity and/or gene regulation) may not be a key factor in the adaptation of microbial communities to different temperatures. Considering that physiological plasticity may slow down intergeneration adaptation to environmental changes because it retards the spread of adaptive genetic traits by shielding the effect of selection[11,44], our data on the response of the individual enzymes to temperature reinforce the consideration that genetic adaptation, by the selection of enzyme variants[10], may have an important role in shaping thermal plasticity of marine microbial communities.

### Response of bacterial community esterase activity to thermal variability

Over a large geographical scale, we showed that the thermal plasticity of enzymes is driven by MAT (Figs. 1 and 2). However, what happens when the thermal variability differs under similar MATs? Assessing the full environmental variability in aquatic habitats using high-resolution temporal and spatial scales—relevant to the individual organisms—can reveal more complex patterns of environmental selection and their

ecological relevance[24,25,45–47]. In tropical clear shallow waters, for example, temperature has a stable mean value over a long period, but it may vary broadly throughout the day/season[46]. Therefore, communities living in these ecosystems can experience environmental conditions far from the value indicated by the averages[25,26]. In this context, the Red Sea represents a model ecosystem with high coastal ecosystem heterogeneity that results in wide thermal variability[48]. Furthermore, it is one of the warmest seas on Earth, with a marked sea surface temperature seasonality ranging from 22 to 24 °C in February to highs of 30–34 °C in August[48–51]. To evaluate the effect of thermal variability on microbial enzymatic response, we selected three adjacent sites in the Red Sea at ~3, ~25, and ~50 m deep characterized by a similar MAT (ΔMAT, ~2.5 °C; Dunnett's multiple comparison tests, adjusted-$p > 0.05$ in all cases), but experiencing significantly different frequency-distribution of temperature, that is different levels of temperature variability, throughout the year (Levene's test $F_{2,52700} = 7328.2$, $p < 2.2E-16$; Fig. 3a, b; Supplementary Tables S4, S5 and Source Data). In the selected sites, sediments experienced high temperature variability (HTV), with temperatures ranging from 21.6 to 34.4 °C (ΔT, 12.8 °C), intermediate temperature variability (ITV), 24.0 °C–32.8 °C (ΔT, 8.8 °C), and low temperature variability (LTV), 24.0–30.7 °C (ΔT, 6.7 °C; Fig. 3a, b). Notably, HTV sediments were exposed to the broadest range of temperatures, including temperatures higher than 31 °C for 45% of the monitoring period; ITV sediments were exposed to

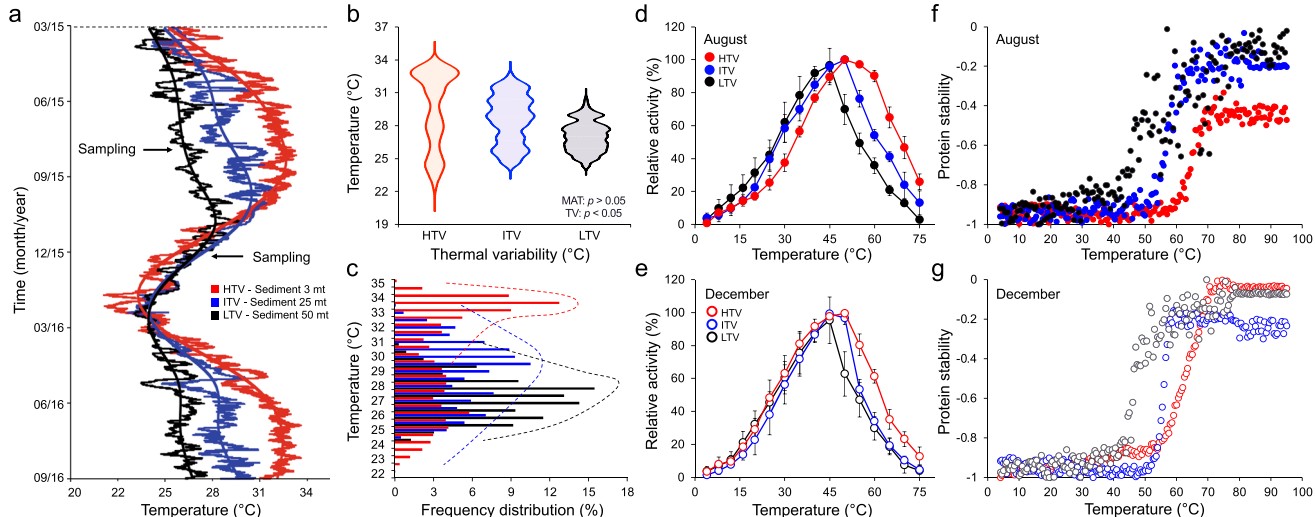

**Fig. 3 | Thermal variability in Red Sea coastal sediments and metabolic response of associated bacterial communities. a** Temperature monitoring over 18 months in high temperature variability (HTV, red), intermediate temperature variability (ITV, blue), and low temperature variability (LTV, black) sediments. Temperatures are expressed as average values from the three stations belonging to each level of thermal variability (see Source Data). Black arrows indicate the sampling sessions. **b** Width of the temperature variability in the sediments from the three stations. The three sites do not show significantly different MAT (Dunnett's multiple comparisons test, all *p*-values <0.05) but significantly different temperature distribution (*i.e.*, temperature variability; Levene's test $F_{2,52700} = 7328.2$, $p < 2.2E\text{-}16$). **c** Frequency distribution (percentages) of temperatures across different levels of temperature variability. **d, e** Activities of esterases extracted from HTV, ITV and LTV sediments. The relative specific activity of the protein extracted from the sediments (units/g of sediment) is expressed as the mean ± SD (*n* = 3 replicates per station per temperature variability, see Source Data) for each temperature tested for the August and December samples, respectively. **f, g** Thermal unfolding patterns of proteins (stability) are measured by CD and expressed as degrees of ellipticity (θ). The normalized CD melting curves of proteins extracted during August and December, respectively, report the θ$_{220}$(T) recorded at 220 nm from 4 °C to 95 °C; data are plotted as the mean of three replicates (see Source Data).

temperatures below 28 °C and between 28 °C and 31 °C for 45 and 43% of the period, respectively, with only 12% of the period at temperatures above 31 °C; LTV sediments were exposed to temperatures below 28 °C for 87% of the period, between 28 °C and 31 °C for 13% of the period, and were never exposed to temperatures above 31 °C (Fig. 3c; Supplementary Table S4). On the contrary, no significant changes in salinity or pH (ANOVA: $F_{2,15} = 1.74$, $p = 0.21$ and $F_{2,15} = 0.32$, $p = 0.73$, respectively; Supplementary Table S5) were observed among the sediments sampled. Furthermore, limited variability in hydrostatic pressure (1–6 atm) and light received at the different sampling depths was observed (data from Bio-ORACLE[52]; Supplementary Table S5). Overall, this suggests that these environmental factors have limited effects on the selection of enzyme properties by the sediment microbial cells[53].

When environmental conditions are variable, such as temperature variability in the considered sediments, the properties of the microbial community are constrained, leading to physiological adaptation and species plasticity[9,54]. We analysed the thermal properties of the whole sediment microbial community proteome (Supplementary Fig. S9) by testing the performance of the esterases present in the active proteomes extracted from HTV, ITV, and LTV samples in August and December under different thermal conditions. We focused on this enzyme class because it has been found to show thermal responses analogous to other enzyme classes (Fig. 1). At the highest temperature tested, the HTV sediment proteomes measured the highest esterase activities both in summer and winter (August and December, Fig. 3d, e), supporting a thermal acclimation of the hydrolytic enzymes to higher and more variable temperatures. Considering the three levels of temperature variability, a statistically significant difference in the $T_d$ of total active/native proteomes (all proteins extracted, including esterases) measured by circular dichroism (CD) was observed in the two seasons (GAM; $F_{2,1096} = 456.2$, $p < 0.0001$ in August, and $F_{2,1096} = 23.89$, $p < 0.0001$ in December), indicating consistently higher stability of proteins (including the enzymes herein examined) extracted from HTV sediments compared to those extracted from ITV and LTV sediments (Fig. 3f, g). Notably, we observed a reduction in the temperatures at which active proteins extracted from HTV and ITV sediments started to denature between the two seasons (Δ$T_d$ of 2.35 °C ± 0.26 °C and 1.12 °C ± 0.23 °C, respectively), while no differences among seasons were detected for proteins from LTV sediments (Fig. 3f, g). The higher stability of the enzymes from the HTV sediments, regardless of the season, supports their higher $T_{opt}$ (measured as esterase activity) of up to 50–60 °C (max. at 51.2 ± 0.3 °C in August and 44.41 ± 0.3 °C in December), compared to enzymes in LTV and ITV sediments that reached their maximum at 40–45 °C (LTV; max. at 45.5 ± 0.4 °C and 43.1 ± 0.5 °C) and 45–50 °C (ITV; max. at 41.1 °C ± 0.4 °C and 40.9 °C ± 0.5 °C; Fig. 3d, e; August, GAM: $F_{2,95} = 8.23$, $p < 0.001$; December, GAM: $F_{2,95} = 4.03$, $p < 0.01$). The enzymatic response was higher in microbial communities that had previously experienced a wider temperature variability and prolonged heat exposure.

These data show that the enzymatic machinery plasticity of microbial communities from marine sediments is linked to the temperature variability of the environment, indicating that the thermal legacy influences microbial community assembly[45,55,56] by selecting microorganisms with more adapted enzymes. For instance, the microbial community enzymes of HTV sediments had a more plastic thermal response than those exposed to less variable environments, namely the ITV and LTV sediments. HTV enzymes showed a higher enzymatic response to high temperatures in both August and December, even though the HTV sediments experienced colder temperatures in winter than the other sediments (Fig. 3a–c). Our data suggest that the thermal legacy of HTV sediments selects more plastic enzymes than those in ITV and LTV sediments experiencing a more stable thermal legacy, which can easily face temperature changes, consistently showing higher activity and stability at high temperatures (e.g., above 45 °C; future temperatures that may be expected in the

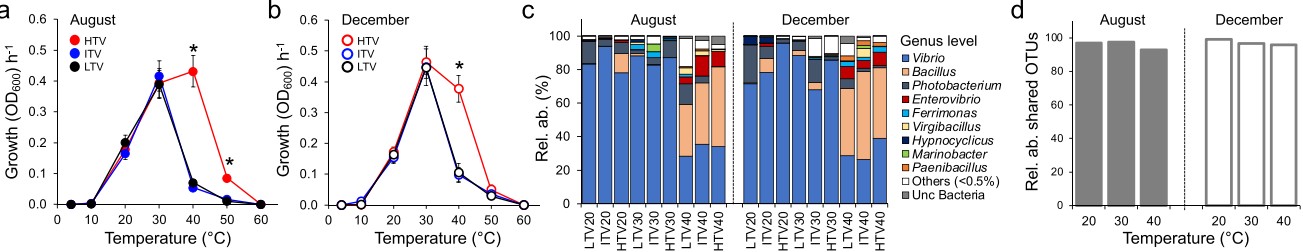

**Fig. 4 | Response of the enriched heterotrophic bacterial fraction from different levels of temperature variability. a**, **b** Growth rates (measured as change in turbidity [$OD_{600}$] per hour) of bacteria from HTV, ITV and LTV sediments sampled in August and December, respectively, measured at 10 °C, 20 °C, 30 °C, 40 °C, and 50 °C. Star (*) indicates statistical differences in the growth of the enriched heterotrophic bacterial fraction obtained from different temperature variabilities across the tested temperatures; ANOVA, **a** August 40 °C: $F_{2,6} = 169.3$, $p < 0.0001$ and 50 °C: $F_{2,6} = 11.04$, $p = 0.0098$, **b** December 40 °C: $F_{2,6} = 23.06$, $p = 0.0015$.

**c** Taxonomic diversity, at the genus level, of the enriched heterotrophic bacteria present at the end of incubation at different temperatures (20 °C, 30 °C, and 40 °C) in August and December; values are expressed as the relative abundance of each genus measured at the end of the incubation (when samples were collected). **d** Relative abundance of OTUs shared across the enriched heterotrophic bacterial community fraction from HTV, ITV and LTV, grown at 20 °C, 30 °C, and 40 °C; values are expressed as percentages.

## Different temperature variability levels in coastal marine sediments are not primarily reflected in the taxonomic compositional turnover of the bacterial communities

The thermal legacy sums up the evolutionary history and, importantly, the thermal events experienced by microorganisms and their enzymes, which contribute to shape their acclimation to temperature variability. We specifically asked if the thermal plasticity of the enzymes of microbial communities experiencing different levels of temperature variability is associated with the compositional turnover of the community.

Species compositional diversity analysis revealed that the different esterase thermal profiles of the microbial communities inhabiting the three sediment groups (LTV, ITV, and HTV) were accompanied by different distributions of bacterial operational taxonomic units (OTUs) in both sampling seasons (Supplementary Table S6). The similarity among the sediment bacterial communities significantly decreased as the temperature difference (ΔT°C) among sediments increased ($p < 0.001$, $R^2 = 0.25$, $n = 1,431$; Supplementary Fig. S10a), with a negative correlation of bacterial richness and evenness with the increment of temperature (Supplementary Fig. S10b). The spatial isolation of sediments and the limited dispersal of their microorganisms reduce the possibility of the introduction of new members into the community[27,59], thus maintaining their differences[60]. Even though certain bacterial species presented significant ($p < 0.05$) enrichment/depletion as a function of sediment temperature (positive correlation, $n = 50$ OTUs and negative correlation, $n = 1041$ OTUs; Supplementary Note S2 and Supplementary Data S5), the majority of the OTUs (82%) were thermal-generalists, and up to 83% were found under all three levels of temperature variability (i.e., shared OTUs in Supplementary Data S5).

To further explore the relationship between microbial diversity and temperature variability, we focused on a subgroup of the microbial communities, the enriched heterotrophic bacteria. We used these microbial subgroups from sediments exposed to LTV, ITV, and HTV, to examine the relationship between growth at different temperatures and microbial diversity with temperature variability. Analogous to observations for the thermal properties of the esterases extracted from the sediments, we detected a significant effect of temperature variability on bacterial growth (measured as optical density, $OD_{600}$) in both seasons (August, GAM: $F_{2,20} = 3.59$, $p < 0.05$; December, GAM: $F_{2,20} = 3.12$, $p < 0.05$; Fig. 4a, b; Supplementary Fig. S11). The bacterial cells extracted from HTV sediments consistently exhibited higher growth rates and ODs at higher temperatures (e.g., 40 °C) than those

from ITV and LTV sediments. Despite the limitation of the cultivation approach, which examines a limited portion of the sediment microbial diversity, it contributes to evaluate if growth performances are driven by a small or large group of taxa. The identification of enriched heterotrophic bacterial taxa from the sediments with different levels of temperature variability and grown at different temperatures revealed the presence of a total of 257 OTUs, mainly affiliated to four genera (Fig. 4c). Members of *Vibrio* were the most abundant (55 OTUs and 69% of relative abundance), followed by *Bacillus* (43 OTUs, 13%), *Photobacterium* (20 OTUs, 7%) and *Enterovibrio* (4 OTUs, 2%). The incubation temperature affected the taxonomic composition of enriched heterotrophic bacteria, but with no major changes across the different temperature variability levels (Fig. 4c; Supplementary Table S7a). Considering the three incubation temperatures separately (20 °C, 30 °C, and 40 °C) in the two sampling seasons, we saw that the enriched heterotrophic bacterial taxa from HTV, ITV and LTV were similar (PERMANOVA, Supplementary Table S7b) with ranges of shared OTUs of 93–98% and 96–99% in August and December, respectively (Fig. 4d). Members of the *Bacillus* genus, for example, were consistently enriched in all three sediments (HTV, ITV and LTV) at high temperature (40 °C). However, only the enriched heterotrophic bacterial taxa from HTV sediments grew at high OD levels (up to 8.3; Supplementary Fig. S11). This indicates that, despite the presence of the same taxa in the sediments, only the bacterial cells with enzymes selected by HTV (Fig. 3d, e) can cope with high temperatures[42,61]. We interpret the finding that the enriched heterotrophic bacterial portion of the bacterial communities exposed to different levels of temperature variability present different growth rates but the same diversity as acclimation of the microbial community to temperature variability rather than a community compositional change.

A broader thermal variability in the thermal legacy of marine sediments selects more plastic microbial communities with a more flexible esterolytic metabolism in response to temperature variability[62,63]. The effect of temperature variability on the thermal plasticity of microorganisms and their enzymes appears as a nested effect over the general thermal effect dictated by MAT. Temperature variability extends and refines the response of the enzymes and the microbial cells to MAT by expanding their thermal plasticity as the temperature variability expands. The discovery of effects of temperature variability on the thermal response of esterases and bacterial growth, even among sediments where temperature variability differences are relatively small such as in the three locations we have examined in the Red Sea, suggests that temperature variability fine-tunes the thermal acclimation of the sediment microbiomes and their enzymes. We speculate that the differences in thermal response observed under the relatively narrow temperature variability

differences measured in the Red Sea could be amplified in other locations with more pronounced seasonality, such as those at intermediate latitudes[64]. Having observed a limited species turnover in response to temperature variability changes, we reason that the effect of temperature variability on the thermal plasticity of microorganisms and their enzymes is, similarly to the effect of MAT, mainly dictated by genetic adaptation and the selection of enzyme variants[10].

In conclusion, our results first showed that microorganisms better adapt to different thermal conditions by selecting thermally-adapted enzymes (through genetic adaptation, e.g., new genes) rather than selecting enzymes with broad thermal tolerances. However, the temperature-dependent expression of isozymes, such as increasing the concentration of enzymes for which reaction rates tend to become temperature-independent and thus work sub-optimally under a new condition (physiological plasticity and/or regulation of genes), could not be ruled out. At the same time, microbial communities living in environments with a wide temperature variability have higher thermal plasticity, in consonance with the thermal legacy of their enzymes. It indicates that the fine-tuned response of microbial communities to temperature change is controlled, at least in part, by the selection of enzyme variants that are capable of being active under more variable temperatures.

The present study illustrates the necessity of investigating the realistic thermal legacy of organisms to accurately interpret their responses to changes in climatic patterns and to evaluate whether this is a planetary-scale mechanism. While the adaptation of microbial communities to temperature change appears to be mediated by the selection of thermally adapted enzymes, the extent of this adaptation also depends on temperature variability, and this should be considered when modelling the response of marine/coastal microbiota to climate change. The incorporation of the recent thermal legacy of microbial communities could, therefore, improve the accuracy of estimates of individual thermal response and hence increase the reliability of predictions on how climate change will affect the assembly, metabolism, and nutrient cycling of the ocean microbiome in the future. Projections between 1950 and 2090 suggest that more than 85% of ecosystems and their microbes will be influenced either directly or indirectly by climate change[65], with the gain or loss of certain groups of bacteria depending on the scenario of fossil-fueled development and its climate effects. In connection to this, our study may suggest that the actual thermal legacy of communities should also be considered to evaluate which microorganisms would be affected by climate change, as well as could influence climate change in a scenario of temperature increase. This will be of considerable interest as the metabolic health and rate of microbiomes is crucial for regulating climate change[66].

## Methods

### Extraction of total active proteomes from sediment samples

We sampled 14 sediments along the coastlines of the Irish Sea, the Mediterranean Sea, and the Red Sea (from 16°N to 53°N), applying uniform sampling and storage procedures. Location details and sediment temperature fluctuations are summarized in Supplementary Table S1. We collected sediments (5 Kg) in triplicate and extracted the total proteins using a well-established microbial detachment procedure[67], with some modifications. We mixed 100 g of sediment with 300 ml of sterilized saline solution (5 mM sodium pyrophosphate and 35 g L$^{-1}$ of NaCl) containing 150 mg L$^{-1}$ of Tween 80 (from Merck Life Science S.L.U., Madrid, Spain) in an ice water bath. After re-suspension, samples were kept in a water bath ultra-sonicator (Bandelin SONOREX, Berlin, Germany) on ice and sonicated (60 W output) for 120 min. We repeated this procedure twice, with an ice water bath incubation of 60 min between each cycle. We then centrifuged the samples at 500 g for 15 min at 4 °C to remove the sediments in a centrifuge 5810 R (Eppendorf AG, Hamburg, Germany). Supernatants were

carefully transferred to a new tube, minimizing disruption of the sediments, and the resulting supernatants were centrifuged at 13,000 g for 15 min at 4 °C to produce microbial cell pellets. We used the resulting cell mix to extract the total protein by mixing the cells with 1.2 ml BugBuster® Protein Extraction Reagent (Novagen, Darmstadt, Germany) for 30 min with shaking (250 rpm). Subsequently, samples were disrupted by sonication using a pin Sonicator® 3000 (Misonix, New Highway Farmingdale, NY, USA) for a total time of 2 min (10 watts) on ice (4 cycles × 0.5 min with 1 min ice-cooling between each cycle). Extracts were centrifuged for 10 min at 12,000 g at 4 °C to separate cellular debris and intact cells. Supernatants were carefully aspirated (to avoid disturbing the pellet), transferred to new tubes, and stored at −80 °C until use. The protein solution was filtered at 15 °C for 7 h using Vivaspin filters (Sartorius, Goettingen, Germany) with a molecular weight (MW) cut-off of 3,000 Da to concentrate the proteins up to a final concentration of 10 mg ml$^{-1}$, according to the Bradford Protein Assay (Bio-Rad Laboratories, S.A., Madrid, Spain)[68]. The average total amount of proteins extracted per each 100 g of sediment was 612 μg (interquartile range, 31 μg, see details in Supplementary Fig. S2). In all cases, extensive dialysis of protein solutions against 40 mM 4-(2-hydroxyethyl)-1-piperazineethanesulfonic acid (HEPES) buffer was performed using a Pur-A-LyzerTM Maxi 1200 dialysis kit (Merck Life Science S.L.U., Madrid, Spain)[69], and active proteins stored at a concentration of 10 mg ml$^{-1}$at −86 °C until use. As reported previously[70], 2DE was performed using GE Healthcare reagents and equipment, 11 cm IPG strips in the pH range of 3–10 and molecular weight ranging from 10 to 250 kDa (Precision Plus Protein Dual Color Standards #1610374, Bio-Rad Laboratories, S.A., Madrid, Spain). The 2-DE was performed using a validated pooling strategy[71], in which proteins extracted from three independent biological replicates (i.e., sediments) were mixed in equal amounts and a total of 150 μg of protein were further loaded per gel. Staining was performed with SYPRO Ruby Protein Gel Stain (Invitrogen, Waltham, MA, USA). The two-dimensional SDS-PAGE (12% acrylamide) gels of extracted proteins are reported in Supplementary Fig. S2 (original gels in Source Data). The same protocol was applied to extract and analyse by SDS-PAGE the total active proteins extracted from sediment samples with different temperature variability levels (HTV, ITV, and LTV) collected in the Red Sea (Supplementary Table S4). The total amount of protein extracted per each 100 g of sediment is given in Supplementary Table S8. Coomassie-stained one-dimension SDS-PAGE (1-DE) gels of extracted proteins are shown in Supplementary Fig. S9 (original gel in Source Data).

### Source, expression and purification of esterases and EXDOs from a wide geographical range

We recovered 83 enzymes (78 esterases and 5 EXDO) from microbial communities inhabiting marine sediments across ten distinct locations from the latitudinal transect described above: Ancona harbour (Anc), Priolo Gargallo (Pri), Gulf of Genoa, Messina harbour (Mes), Milazo harbour (Mil), Mar Chica lagoon (MCh), Bizerte lagoon (Biz), El-Max site (ElMax), Gulf of Aqaba (Aq), and Menai Strait (MS); further details are provided in Supplementary Data S3. Sources of the enzymes were the corresponding shotgun metagenomes (see Supplementary Table S3) and the metagenome clone libraries generated from the extracted DNA[71]. The sediment sample from the Gulf of Genoa was not used for activity tests and metaproteome analysis because no raw sample material was available; however, because of the possibility to access its shotgun metagenome (see Supplementary Table S3) and a metagenome clone library[72], we used the sample for screening esterases to incorporate an additional latitude in our transect. In the case of Menai Strait (Irish Sea), five additional esterases were retrieved from a metagenome obtained from enriched cultures prepared with samples collected on 22$^{nd}$ June 2019 from Menai Strait (School of Ocean Sciences, Bangor University, St. George's Pier, Menai Bridge,

N53°13′31.3″; W4°09′33.3″). The water temperature was 14 °C and the salinity was 32 p.s.u. Two enrichment cultures were set up at 20 °C: (i) SW: seawater enrichment with 0.1% lignin; the enrichment was set up using 50 ml of the sample as inoculum with the addition of 0.1% lignin (Sigma-Aldrich, Gillingham, United Kingdom) (w/v); (ii) AW: algal surface wash-off in seawater, enriched with 0.1% lignin; the enrichment was set up using 50 ml of surface wash-off after mixing of ca. 10 g of Fucus (brown algae) in the seawater and removal of plant tissue, 0.1% lignin (w/v) was added. After 92 days of incubation, 5 ml of each enrichment cultures were transferred into the new flask containing 45 ml autoclaved and filtered seawater with 0.1% lignin. This procedure was repeated on days 185 and 260, and the incubation was stopped on day 365. The DNA was extracted using 12 months using MetaGnome extraction kit (EpiCentre, Biotechnologies, Madison, WI, USA), sequenced on Illumina MiSeq™ platform (Illumina Inc., San Diego, CA, USA) using paired-end 250 bp reads at the Centre for Environmental Biotechnology (Bangor, UK), and sequencing reads were processed and analysed as described previously[73].

The screening, cloning and activity of a subset of 35 identified esterases have been reported previously[72]. The remaining 48 enzymes are reported for the first time in this study and were identified using naive and in silico metagenomic approaches, as detailed below. The environmental site from which each enzyme originated and the method employed for its identification are detailed in Supplementary Data S3. For naive screens addressing the recovery of new sequences encoding esterases and EXDO, the large-insert pCCFOS1 fosmid libraries made using the corresponding DNA samples, the Copy-Control Fosmid Library Kit (Epicentre Biotechnologies, Madison, WI, USA) and the *Escherichia coli* EPI300-T1R strain were used. The nucleic acid extraction, construction and the functional screens of such libraries have been previously described[72]. In brief, fosmid clones were plated onto large (22.5 × 22.5 cm) Petri plates with Luria Bertani (LB) agar containing chloramphenicol (12.5 µg ml⁻¹) and induction solution (Epicentre Biotechnologies; WI, USA), at a quantity recommended by the supplier to induce a high fosmid copy number. Clones were scored by the ability to hydrolyze α-naphthyl acetate and tributyrin (for esterase activity), and catechol (for EXDO activity)[72,74]. Positive clones presumed to contain esterases and EXDOs were selected, and their DNA inserts were sequenced using a MiSeq Sequencing System (Illumina, San Diego, USA) with a 2 × 150-bp sequencing v2 kit at Lifesequencing S.L. (Valencia, Spain). After sequencing, the reads were quality-filtered and assembled to generate nonredundant metasequences, and genes were predicted and annotated via BLASTP and the PSI-BLAST tool[72]. For in silico screens, addressing the recovery of new sequences encoding esterases, the predicted protein-coding genes, obtained after the sequencing of DNA material from resident microbial communities in each of the samples, were used. The metasequences are available from the National Center for Biotechnology Information (NCBI) nonredundant public database (accession numbers reported in Supplementary Data S3). Protein-coding genes identified from the DNA inserts of positive clones (naive screen) or from the meta-sequences were screened for enzymes of interest using the Blastp algorithm via the DIAMOND v2.0.9 program with default parameters (percentage of identity ≥60%; alignment length ≥70; e-value ≤1e⁻⁵)[29], against the Lipase Engineering sequence databases (to screen for esterases) and AromaDeg database (for EXDO)[74]. Since the collection of sediments across locations experiencing different MATs was limited by our sampling capacity, to expand our range of exploration at a global scale and to validate our dataset, we added our single enzyme analysis to the seawater metagenomes retrieved from the Tara Ocean Expedition database (accession number in Supplementary Data S4). Due to the volume of sequences generated, this database provides access to a large number of enzymes, including those studied here through homology search. Esterases were selected as target sequences, and the following pipeline was used. First, we selected a

sequence encoding an esterase reported as one of the most substrate-ambiguous esterases out of 145 tested (EH₁, Protein Data Bank acc. nr. 5JD4) and well-distributed in the marine environment[72]. Second, we performed a homology search of this sequence against the Tara Ocean metagenome[21] to retrieve similar sequences, using the Blastp algorithm via the DIAMOND v2.0.9 program[30] (e-value <e⁻¹⁰). A total of 150 sequences encoding presumptive such enzymes from 56 different locations of the Tara Ocean Expedition were selected (Supplementary Data S4).

Once identified, the sequences encoding the wild-type enzymes—here identified and reported for the first time from all the geographically distinct locations (including the ones from the Tara Ocean Expedition)—were used as templates for gene synthesis. Genes were codon-optimized to maximize expression in *E. coli*. Genes were flanked by *Bam*HI and *Hind*III (stop codon) restriction sites and inserted in a pET-45b(+) expression vector with an ampicillin selection marker (GenScript Biotech, EG Rijswijk, Netherlands). This plasmid, which was introduced into *E. coli* BL21(DE3), supports the expression of N-terminal histidine (His) fusion proteins with the final amino acid sequences of all synthetic proteins being MAHHHHHHVGTGSNDD DDKSPDP-X, where X corresponds to the original sequence of the target enzyme (Supplementary Data S3 and S4). In all cases, the soluble His-tagged proteins were produced and purified at 4 °C after binding to a Ni-NTA His-Bind resin (Merck Life Science S.L.U., Madrid, Spain), as described previously[72,74]. Purity was assessed as > 98% using SDS-PAGE analysis in a Mini PROTEAN electrophoresis system (Bio-Rad Laboratories, S.A., Madrid, Spain). Purified protein was stored at −86 °C until use at a concentration of 10 mg ml⁻¹ in 40 mM HEPES buffer (pH 7.0). A total of approximately 5–40 mg of total purified recombinant protein was obtained from 1 L of culture. Supplementary Fig. S1 illustrates a schematic representation of the pipeline implemented in this work to investigate enzyme activities in a large set of marine samples, starting from samples collected (sediments) and available metagenomes.

## Enzyme activity assessments

All substrates used for activity tests were of the highest purity and, if not indicated otherwise, were obtained from Merck Life Science S.L.U. (Madrid, Spain): 4-nitrophenyl-propionate (ref. MFCD00024664), 4-nitrophenyl phosphate (ref. 487663), 4-nitrophenyl β-D-galactose (ref. N1252), bis(p-nitrophenyl) phosphate (ref. 123943), benzaldehyde (ref. B1334), 2-(4-nitrophenyl)ethan-1-amine (ref. 184802-5G), pyridoxal phosphate (ref. P9255), acetophenone (ref. A10701), NADPH (ref. N5130) and catechol (ref. PHL82372). We directly tested total protein extracts for esterase, phosphatase, beta-galactosidase, and nuclease activity using 4-nitrophenyl-propionate, 4-nitrophenyl phosphate, 4-nitrophenyl β-D-galactose, and bis(p-nitrophenyl) phosphate, respectively, by following the production of 4-nitrophenol at 348 nm (extinction coefficient [ε], 4147 M⁻¹ cm⁻¹), as previously described[69]. For determination: [total protein]: 5 µg ml⁻¹; [substrate]: 0.8 mM; reaction volume: 200 µl; T: 4–85 °C; and pH: 8.0 (50 mM Tris-HCl buffer). The hydrolysis of 4-nitrophenyl-propionate was used to determine, under these standard conditions, the effects of temperature on the purified esterase. Transaminase activity was determined using benzaldehyde as amine acceptor, 2-(4-nitrophenyl)ethan-1-amine as amine donor, and pyridoxal phosphate as a cofactor, by following the production of a colour amine at 600 nm (extinction coefficient, 537 M⁻¹ cm⁻¹), as previously described[75]. For determination, [total protein]: 5 µg ml⁻¹; [substrates]: 25 mM; [pyridoxal phosphate]: 1 mM; reaction volume: 200 µL; T: 4-85 °C; and pH: 8.0 (50 mM Tris-HCl buffer). Aldo-keto reductase activity was determined using acetophenone as a substrate and NADPH as a cofactor, by following the consumption of NADPH at 340 nm (extinction coefficient, 6220 M⁻¹ cm⁻¹), as described[76]. For determination, [total protein]: 5 µg ml⁻¹; [substrate]: 1 mM; [cofactor]: 1 mM; reaction volume: 200 µL; T: 4–85 °C; and pH: 8.0 (50 mM Tris-HCl buffer). We determined EXDO activity using catechol as substrate,

by following the increase of absorbance at 375 nm of the ring fission products (extinction coefficient, 36000 $M^{-1} cm^{-1}$), as previously described[74]. For determination, [protein]: 5 µg ml⁻¹; [catechol]: 0.5 mM; reaction volume: 200 µL; T: 4–85 °C; and pH: 8.0 (50 mM Tris-HCl buffer). The hydrolysis of catechol was used to determine, under these standard conditions, the effects of temperature on the purified EXDOs. All measurements were performed in 96-well plates (ref. 655801, Greiner Bio-One GmbH, Kremsmünster, Austria), in biological triplicates over 180 min in a Synergy HT Multi-Mode Microplate Reader (Biotek Instruments, Winooski, VT, USA) in continuous mode (measurements every 30 s) and determining the absorbance per minute from the slopes generated and applying the formula (1). All values were corrected for nonenzymatic transformation.

$$Rate\left(\frac{\mu mol}{min \, mg \, protein}\right) = \frac{\frac{\triangle Abs}{min}}{\varepsilon, M - 1cm - 1} * \frac{1}{0.4 \, cm} * \frac{10^6 \, \mu M}{1M}$$
$$*0.0002 \, L * \frac{1}{mg \, protein} \tag{1}$$

## Shotgun proteomics

Proteomics was performed by using total active proteins (extracted as above), which were then subjected to protein precipitation, protein digestion and Liquid Chromatography-Electrospray Ionization Tandem Mass Spectrometric (LC-ESI-MS/MS) analysis, as previously described[77]. High-quality reference metagenomes corresponding to each sample (BioProject number in Supplementary Table S3) were used for protein calling, with a threshold of only one identified peptide per protein identification because False Discovery Rates (FDR) controlled experiments counter-intuitively suffer from the two-peptide rule. The confidence interval for protein identification was set to ≥95% ($p < 0.05$), and only peptides with an individual ion score above 20 were considered correctly identified. All protocols and experimental details, including those for mass spectrometry and measures of quality and fidelity of the datasets, have been previously described[77]. The mass spectrometry proteomics data have been deposited to the ProteomeXchange Consortium via the PRIDE[78] partner repository with the dataset identifier PXD039714 and 10.6019/PXD039714.

## Thermal denaturation assessments of proteins through circular dichroism

Spectra were acquired between 190 and 270 nm with a Jasco J-720 spectropolarimeter equipped with a Peltier temperature controller, employing a 0.1 mm cell at 25 °C. Spectra were analysed, and $T_d$ values were determined at 220 nm between 10 °C and 85 °C at a rate of 30 °C per hour in 50 mM Britton and Robinson buffer at pH 8.5. A protein concentration of 1.0 mg ml⁻¹ was used. $T_d$ (and standard deviation of the linear fit) was calculated by fitting the ellipticity (millidegrees; θ) at 220 nm at each of the different temperatures using a 5-parameter sigmoid fit with SigmaPlot 13.0[79,80]. We used a generalized additive model (GAM)[81] to analyse $T_d$ (our response variable) at each temperature (continuous explanatory variable) in the hot and cold seasons (in August and December, respectively) and the three levels of temperature variability (HTV, LTV, ITV) as our two categorical explanatory variables.

## Enzyme structure prediction, ensemble generation and rigidity analysis

We applied the AlphaFold2-based workflow of ColabFold[82,83], which is available at https://github.com/sokrypton/ColabFold (accessed 22.02.2022), to generate 3D structural models of esterases. A single model was generated for each esterase with ten prediction cycles (--num_recycles) and structurally refined by running a relaxation with AMBER (--amber). For subsequent analyses, only esterases with a sufficient 3D structural model quality, with a sequence length <1000

residues, and without cofactors were considered. To test whether the catalytically active residues (CARs) of the 3D structures are accessible for substrates, we used the CAVER 3.0.3 PyMOL Plugin[84,85]. Therefore, CARs were identified based on the minimal summed distances between all triplets of serine oxygen atoms, histidine nitrogen atoms in epsilon position, and carbon atoms of the carboxylic acid group from either glutamate or aspartate residues. Starting points for the computations were defined based on the Cartesian coordinates of the centre of mass (COM) of each CAR. Default values were used for the probe radius (0.9 Å), shell radius (3.0 Å) and shell depth (4.0 Å). We verified that CARs in all models are accessible for substrates, i.e., that all models are in an open conformation: CARs are either located on the protein surface or are buried and connected with the surface by tunnels.

The esterase structures were pre-processed with pdb4amber, which is part of AmberTools21, and hydrogen atoms were added using the Reduce program[86]. The prepared esterases were solvated in a truncated octahedron of TIP3P water[87], leaving at least 20 Å between the esterase structure and the edges of the solvent box, using the LeaP program of AmberTools21. All systems were neutralized by adding $Na^+$ or $Cl^-$ ions as needed. We used the Amber ff14SB force field[88] to parametrize the protein. Ion parameters were taken from Joung and Cheatham[89]. Structural ensembles of esterases were generated by all-atom molecular dynamics (MD) simulations, with five replicas at 500 ns, yielding 2.5 µs of cumulative simulation time per esterase. Minimization steps, thermalization, and production simulations were carried out using the GPU-accelerated CUDA version of PMEMD[90,91] from the Amber21 suite of programs[92]. The systems were heated to 298 K and the pressure was adapted in NPT simulations to obtain a density of 1 g cm⁻³. During thermalization and density adaptation, we kept the solute fixed by positional restraints of 1 kcal mol⁻¹ Å⁻², which were gradually removed over five steps in short subsequent NVT simulations. Afterwards, five NVT production simulations of 500 ns length were performed using unbiased MD simulations. During these simulations, we set the time step to integrate Newton's equation of motion to 4 fs, following the hydrogen mass repartitioning strategy[93]. Coordinates were saved every 200 ps yielding 2500 conformations for each production run that were considered for subsequent analyses. The Amber21 software for MD simulations is available at http://ambermd.org/.

Rigidity analyses were performed following Nutschel et al.[41] using the Constraint Network Analysis (CNA) software package (version 3.0)[33,34,94,95], available at https://cpclab.uni-duesseldorf.de/index.php/Software. In detail, we applied CNA on ensembles of network topologies generated from conformational ensembles obtained from MD simulations generated at room temperature, for which force field parameters had been optimized. Average stability characteristics were calculated by constraint counting on each topology in the ensemble. CNA functions as a front- and back-end to the graph theory-based software Floppy Inclusions and Rigid Substructure Topography (FIRST)[96]. The application of CNA to biomolecules aims to identify their rigid cluster and flexible region composition, which can aid in understanding the biomolecular structure, stability and function. As the mechanical heterogeneity of biomolecular structures is intimately linked to their diverse biological functions, biomolecules generally show a hierarchy of rigid and flexible regions[37]. To monitor this hierarchy, CNA performs thermal unfolding simulations by consecutively removing noncovalent constraints (hydrogen bonds and salt bridges) from a network in the order of their increasing strength. Therefore, a hydrogen bond energy $E_{HB}$ is computed from an empirical energy function[97]. For a given network state $\sigma = f(T)$, hydrogen bonds (including salt bridges) with an energy $E_{HB} > E_{cut}(\sigma)$ are removed from the network at temperature $T$. In the present study, thermal unfolding simulations were carried out by reducing $E_{cut}$ from −0.1 kcal mol⁻¹ to −6.0 kcal mol⁻¹ with a step size of 0.1 kcal mol⁻¹. $E_{cut}$ can be converted

to a temperature $T$ using the linear equation introduced by Radestock et al.[33] as reported in equation (2), where the range of $E_{cut}$ is equivalent to increasing temperature from 302 K to 420 K with a step size of 2 K.

$$T = -\frac{20K}{\left(kcal\ mol^{-1}\right)}*E_{cut} + 300K \qquad (2)$$

The number of hydrophobic tethers was kept constant during the thermal unfolding simulations[98]. From these simulations, CNA computes a set of indices to quantify biologically relevant characteristics of protein stability at a global and local scale[34,99]. Here, we used the cluster configuration entropy $H_{type2}$, a measure of the global structural stability, to predict the phase transition temperature $T_p$ (for details on $H_{type2}$[34,37,99]). At $T_p$, the protein switches from a rigid (structurally stable) to a floppy (unfolded) state and the largest rigid cluster no longer dominates the whole protein network. If the largest rigid cluster dominates the whole protein network, $H_{type2}$ is low because of the limited number of possible ways to configure a system with a very large cluster. When the largest rigid cluster starts to decay or stops to dominate the network, $H_{type2}$ jumps. At this stage, the network is in a partially flexible state with many ways to configure a system consisting of many small clusters. The percolation behaviour of protein networks is usually complex, and multiple phase transitions can be observed[33,35,37–41,100]. To identify $T_p$, a double sigmoid fit was applied to an $H_{type2}$ *versus* $T(E_{cut})$ curve as performed previously[33,35,37–41,100]. In general, $T_p$ was taken as the $T$ value associated with the largest slope of the fit, except for esterases with $T_d > 50\,°C$ for which the second phase transition was chosen to focus on the decomposition of the core. It is important to note that applying CNA to MD simulations at room temperature may lead to an evening out of $T_p$ values for esterases that transition around this temperature, i.e., systems with a $T_p$ at or below room temperature might all be influenced similarly by loosening their bonding network. By contrast, systems with a transition temperature at or above room temperature would still be discriminated against. The data generated in this study for analyzing $T_p$ values have been deposited at researchdata.hhu.de under accession code DOI: 10.25838/d5p-42[101] [https://doi.org/10.25838/d5p-42].

### Relationship of temperature-induced changes in enzyme

Relationship between MAT and enzyme response to temperature (i.e., $T_{opt}$, $T_d$ and $T_p$) were evaluated by performing linear regression in R. In the case of enzymes retrieved from the Tara ocean dataset we calculated first the break point (flexus) using the package *segmented* in R[102] and then we computed separately the linear model describing the two linear regressions before and after the breakpoint. To evaluate the possible relation between enzyme thermal response and other environmental parameters, salinity and pH data were retrieved from Bio-ORACLE[52] using GPS coordinates of each location.

### Environmental characterization and sediment collection from different temperature variability levels in the Red Sea

We recorded the temperatures of surface sediments from March 2015 to September 2016 along the coast of the Red Sea using HOBO data loggers (Onset, USA) in nine stations located at 3, 25, and 50 m depth. Details on the location, depth and temperature fluctuations of the studied sediments are reported in Supplementary Table S4 and Source Data. We first assess the differences in the homogeneity of the temperature variance in the three types of sediments to evaluate the magnitude of thermal variation and then we test the difference among their MATs using a non-parametric ANOVA (Dunnett's multiple comparisons tests). We identified three different levels of temperature variability (Fig. 3a–c; Supplementary Table S5): high, intermediate, and low thermal variability (HTV, ITV, and LTV, respectively), where sediments experienced temperature variations of 12.8 °C, 8.8 °C, and 6.7 °C, respectively. From each station, we sampled 200 g of surface

sediment (0–5 cm depth) in triplicate in August and December 2015 with a Van der Venn grab (1 dm³) equipped with a MicroCat 250 Seabird CTD (Conductivity, Temperature, Depth), which was assembled on board the research vessel R/V Explorer (KAUST). During sampling, we measured the temperature of the sediments and the water layer covering the sediments using a digital thermometer and the CTD, respectively. We conducted all sampling in compliance with the guidelines specified by KAUST and Saudi Arabian authorities.

### Sediment processing for analysis of bacterial communities

From each sample (in triplicate), we immediately removed subsamples of sediment ($n = 54$, ~10 g) and stored them at −20 °C for molecular analysis. Separately, sediment $25 \pm 1\,g$ was transferred to 50 ml tubes and added 30 ml of filtered (0.2 μm) water from the Red Sea. The tubes were shaken at 500 rpm for one hour and then centrifuged them at 300 $g$ for 15 min to detach the microbial cells in the sediments without affecting their vitality[103,104]. The supernatant containing the extracted cells was collected in sterile tubes and was immediately used to measure microbial growth rates.

### Evaluation of bacterial growth in sediments at different temperatures

We evaluated the microbial growth rate of the heterotrophic community extracted from the sediments under HTV, ITV, and LTV at 10 °C, 20 °C, 30 °C, 40 °C and 50 °C, using Marine Broth as the cultivation medium (Zobell Marine Broth 2216) supplemented with 0.1 g/L cycloheximide; a rich-medium was selected to avoid the nutrient limitation effect that can affect bacterial physiology[63,105]. We inoculated 96-well plates with 200 μl of cultivation medium and 25 μl of the cell suspension extracted from the sediments. We inoculated the three biological replicates from each station and each level of temperature variability in eight wells, giving a total of 72 wells for each plate, with 24 wells used as a negative control inoculated with water. We assembled a total of three plates for each incubation temperature from August and December. Plates were spectrophotometrically measured at 3 h intervals using an optical density of 600 nm (Spectramax® M5) for 72 h. Wells with optical density <0.15 (average value of the medium inoculated with autoclaved sediment extracts) were assigned as 'no-growth' at a given temperature. Growth (expressed as $OD_{600}$) was normalised in terms of the $OD_{600}$ of the initial inoculum. The growth rate was calculated as the change in the number of cells in a culture per unit of time (h)[106]. A GAM[81] was applied to evaluate the effect of temperature, temperature variability level and seasonality on the continuous response variables of bacterial growth and growth rate.

### Total DNA extraction, Illumina sequencing, and taxonomic analysis of bacterial 16S rRNA gene sequences

The total DNA was extracted from $0.4 \pm 0.1\,g$ of sediment using a DNeasy PowerSoil Pro Kit® (Qiagen) and from the final enriched heterotrophic bacterial fraction obtained by the cultivation approach using a DNeasy UltraClean Microbial Kit (Qiagen). PCR amplification of the V3–V4 hypervariable regions of the 16S rRNA gene on DNA in the sediment samples was performed using the universal primers 341f and 785r[107]. We constructed libraries with the 96 Nextera XT Index Kit (Illumina) following the manufacturer's instructions and sequenced DNA using the Illumina MiSeq® platform with paired-end sequencing at the Bioscience Core Lab at KAUST. Raw reads were deposited in the NCBI database under the SRA accession number PRJNA508596. We assembled forward and reverse reads for each sample into paired-end reads (minimum overlap of 50 nucleotides and a maximum of one mismatch within the region) using the fastq-join algorithm, and the samples were analysed using the UPARSE v8 and QIIME v1.9 softwares[108]. The final reads (average length of 405 bases) were clustered into operational taxonomic units (OTUs), taking 97% sequence identity as the cut-off. All samples showed a sufficient

sequencing depth (Good's coverage values >90%) for further analysis (Supplementary Tables S9 and S10). We calculated the compositional similarity matrix (Bray-Curtis of the log-transformed OTU table) with Primer 6[109]. Using the same software, canonical analysis of principal coordinates (CAP)[110] was used to compare the temperature variability samples (temperature variability levels: HTV, ITV, and LTV; season levels: August and December) based on the compositional similarity matrix. We applied permutational multivariate analyses of variance to the matrix (PERMANOVA; main and multiple comparison tests). We tested the occurrence of thermal-decay patterns in sediments with different temperature variability levels using linear regression (Prism 9.2 software, La Jolla California USA, www.graphpad.com) between the bacterial community similarities (Bray-Curtis) and the temperature differences among sediments (ΔT°C) at the time of sampling. We calculated alphadiversity indices (richness and evenness) using the paleontological statistics (PAST) software, and their correlation with temperature was modelled using linear regression in Prism 9.2. Spearman correlation among temperature and relative abundance of OTUs within each sediment sample was evaluated; OTUs were classified based on their positive (enriched) and negative (depleted) correlation with sediment temperature.

### Reporting summary

Further information on research design is available in the Nature Portfolio Reporting Summary linked to this article.

## Data availability

The authors declare that the main data supporting the findings of this study are available within the paper and related Supplementary Information, Supplementary Data and Source Data files. Accession numbers to retrieve metagenomes analysed in this study are reported in Supplementary Table S3, Supplementary Data S3 and Supplementary Data S4 files. The mass spectrometry proteomics data are available via ProteomeXchange with identifier PXD039714. The data related to $T_p$ have been deposited at researchdata.hhu.de under the identifier [https://doi.org/10.25838/d5p-42][101]. To use the archive, download the file, remove the .txt ending, and use WinRAR, for example, to open the archive. Microbiome sequences extracted from temperature variability sediments and related enriched heterotrophic bacteria were deposited in the NCBI database under the SRA accession number PRJNA508596. Source data are provided with this paper.

## Code availability

The scripts for analysis are available on GitHub at github.com/MarcoFusi1980/Enzyme-adaptation-to-habitat-thermal-legacy-explains-the-plasticity-of-marine-microbiomes.git.

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

## Acknowledgements

We thank Sadaf Umer and Taskeen Begum for their invaluable support in the lab, the Coastal and Marine Resources Core Lab at KAUST for their support in sampling in the Red Sea and the Bioscience Core Lab at KAUST for their support in DNA sequencing. This research was sup-ported by King Abdullah University of Science and Technology through baseline funding to D.D. We thank our partner in the EU project ULIXES for providing sediment samples from the Mediterranean Sea and the Gulf of Aqaba. This study was conducted under the auspices of the

FuturEnzyme Project funded by the European Union's Horizon 2020 Research and Innovation Programme under Grant Agreement No. 101000327 (acknowledged by M.Fe. and P.N.G.). We also acknowledge financial support under Grants PID2020-112758RB-I00 (M.Fe.), PDC2021-121534-I00 (M.Fe.) and TED2021-130544B-I00 (M.Fe.) from the Ministerio de Ciencia e Innovación, Agencia Estatal de Investigación (AEI) (Digital Object Identifier 10.13039/501100011033), Fondo Europeo de Desarrollo Regional (FEDER) and the European Union ("NextGenerationEU/PRTR"), and Grant 2020AEP061 (M.Fe.) from the Agencia Estatal CSIC. P.N.G. acknowledges the Sêr Cymru programme partly funded by ERDF through the Welsh Government for the support of the project Bio-POL4Life, the project 'Plastic Vectors' funded by the Natural Environment Research Council UK (NERC), Grant No. NE/S004548/N and the Centre for Environmental Biotechnology Project co-funded by the European Regional Development Fund (ERDF) through the Welsh Government. M.Fe. also acknowledges Sergio Ciordia and M. del Carmen Mena, who performed SDS-PAGE and shotgun proteomic analyses at the Proteomics Facility of the Spanish National Center for Biotechnology, ProteoRed, PRB3-ISCIII. Parts of the study were supported by the German Federal Ministry of Education and Research (BMBF) through funding number 031B0837A "LipoBiocat" to H.G. and the state of North-Rhine Westphalia (NRW) and the European Regional Development Fund (EFRE) through funding no. 34-EFRE-0300096 "CLIB-Kompetenzzentrum Biotechnologie (CKB)" to H.G. H.G. is grateful for computational support and infrastructure provided by the "Zentrum für Informations- und Medientechnologie" (ZIM) at the Heinrich Heine University Düsseldorf. H.G. gratefully acknowledges the computing time granted by the John von Neumann Institute for Computing (NIC) and provided on the supercomputer JUWELS at Jülich Supercomputing Centre (JSC) (user IDs: VSK33, lipases).

## Author contributions

D.D. and M.Fe. conceived the original idea of the study; D.D., M.Fe., and P.N.G. wrote the applications for the grants that funded the work; R.Mar., M.Fu., D.D., C.C., and M.Fe. designed the experiments; R.Mar., M.Fu., and A.B. collected sediment samples in the Red Sea, and F.M. and D.D. helped to collect samples in the Mediterranean Sea; P.N.G. provided Irish Sea sediments and contributed to gene cloning; R.Mar., M.Fu., C.C., A.B., D.A., R.Mat., R.B., and S.S.C. performed the laboratory, bioinformatics or computational work a nd analysed data; T.N.C. contributed preparing the Menai straits (Irish Sea) seawater lignin enrichment and sequencing; J.D. performed MD simulations, C.G. validated the structural models and analysed CNA runs, C.P. performed structural modelling, identified CARs, and performed CNA runs, H.G. supervised the modelling and simulation work; R.Mar. and M.Fu. wrote the first version of the manuscript with contributions from C.C., M.Fe and D.D.; All authors contributed to the revision of the manuscript.

## Competing interests

The authors declare no competing interests.

## Additional information

[1]Biological and Environmental Sciences and Engineering Division (BESE), Red Sea Research Centre (RSRC), King Abdullah University of Science and Technology (KAUST), Thuwal, Saudi Arabia. [2]Centre for Conservation and Restoration Science, Edinburgh Napier University Sighthill Campus, Edinburgh, UK. [3]Instituto de Catalisis y Petroleoquimica (ICP), CSIC, Madrid, Spain. [4]Centre for Environmental Biotechnology, School of Natural Sciences, Bangor University, Deiniol Rd, Bangor, UK. [5]Institute of Bio- and Geosciences (IBG-4: Bioinformatics), Forschungszentrum Jülich GmbH, Jülich, Germany. [6]Mathematisch-Naturwissenschaftliche Fakultät, Institut für Pharmazeutische und Medizinische Chemie, Heinrich-Heine-Universität Düsseldorf, Düsseldorf, Germany. [7]John von Neumann Institute for Computing (NIC) and Jülich Supercomputing Centre (JSC), Forschungszentrum Jülich GmbH, Jülich, Germany. [8]Spectroscopy Laboratory, Centro de Investigaciones Biologicas Margarita Salas (CIB), CSIC, Madrid, Spain. [9]Centro de Biologia Molecular Severo Ochoa (CBM), CSIC-UAM, Madrid, Spain. [10]Department of Food Environmental and Nutritional Sciences, University of Milan, Milan, Italy. [11]These authors contributed equally: Ramona Marasco, Marco Fusi, Cristina Coscolín. ✉e-mail: mferrer@icp.csic.es; daniele.daffonchio@kaust.edu.sa

