## [Peer Review File · Nature Communications]

Enzymes' adaptation to habitat thermal legacy explains the plasticity of marine microbiomesReviewers' Comments:

Reviewer #1:

Remarks to the Author:

The study by Marasco et al explores the hypothesis that enzyme thermal selection explains the plasticity of marine microbiomes in response to temperature.

Understanding regulatory mechanisms of thermal stress response in the marine microbiome is a welcomed addition to the broad interests in the thermal plasticity and adaptability of microbiome facing risks of changing environmental conditions. Knowledge of the complexity of adaptations is crucial to infer which species will become winners or losers, and to predict resilience and tipping points. In addition, marine microbiomes have not been scrutinized to the same extent as their terrestrial counterparts so I encourage studies in this direction.

Below I suggest ways by which the authors could improve the interpretation and discussion of their results that might help the reader understand the significance of the data, as well as provide a landmark comparative framework for future studies. I think these minor edits would strength the manuscript but are not necessary for publication and can be done, if desired, at the authors discretion.

My few minor comments are relative to some narrative aspects, putting in a bit more detail in the Introduction, and Results section.

INTRODUCTION:

Perhaps add more nuance to the introduction of the differences between genetic adaptation and physiological plasticity – they are currently not introduced and should be explicitly entwined. Reading the Introduction, it feels like you're slightly underselling the importance that bacteria (short-generation time species) may also have high potential for timely genetic adaptation and produce new variants. The narrative should reconcile this complexity more clearly.

METHODS:

Authors studied the thermal properties of 80 isolated enzymes from 14 sediments along the above mentioned latitudinal transect and 150 additional enzymes retrieved from 56 seawaters locations from Tara Ocean expedition. By quantifying the microbial enzyme temperature-dependent profiles in proteomes of marine sediments from the Irish Sea to the southern Red Sea authors recorded a significant temperature-dependent response of the enzymatic activity.

The lack of general optimised methods of samples collection often produces results that may be controversial and do not indicate the true trend or introduce limits in data interpretation. It would have been more effective and simpler to draw conclusions if more homogenous samples would have been taken and analysed. This issue (bias) also refers to the cloning and activity of ester-hydrolases some analysed in this study and some others retrieved from previous studies. I understand the difficulty in accessing samples at various latitudes in a FAIR way and I acknowledge the significance in publishing these results as they stand.

RESULTS DISCUSSION:

---The paper as written lacks clarity on the the mechanisms of temperature compensation through natural selection (never mentioned) and physiological plasticity that are generally adopted for coping with changes of temperatures. Authors focus their conclusions exclusively on the changes of activity of enzymes as sign of thermal adaptation with no mention on changes in flexibility (the most common response to changing conditions at protein level).

Changes in protein flexibility may play an important role at diverse temperatures without altering the global structure and the active site, thus reducing the selective pressure on new genes.

---Nature always produces cold-adapted proteins with decreased stability and viceversa. In addition, most studies indicate an inverse relationship between stability and activity while in your study it seems that there is a direct correlation between these two properties e.g. higher stability-higher activity.

---Some of the strategies developed at the protein level include (i) increasing of enzyme concentration (physiological plasticity), ii) temperature dependent expression of iso-enzymes (physiological plasticity, regulation of genes), and new variants of enzymes for which reaction rates tend to become temperature independent or adapted to new conditions (genetic adaptation, e. g new genes). Authors have any functional evidence/information on the up- and/or down-regulated expression of genes encoding the enzymes selected in the study?

REFERENCES:

---The few references on protein thermal adaptation are old. More references on this should be included.

Reviewer #2:

Remarks to the Author:

The research led by Marasco et al, studying the thermal tolerance of xylase hydrolytic activity across a wide variety of enzyme preparations from marine sediments and water columns differing in temperature was a large-scale effort aimed at uncovering underlying patterns in thermal adaptation and tolerance range. The work is novel through applying a broad biogeographic approach to enzyme thermal property characterizations, where it incorporates an ecological perspective that considers the thermal legacy of a site (and organisms therein) to interpret the thermal tolerance of the hydrolytic enzymes.

Key results demonstrated that (i) enzymes/proteomes exhibit a temperature dependent range of hydrolytic esterase activity that relates to their native habitat thermal properties, (ii) esterase activity from sites with different thermal variance ranges (low - intermediate - high) show different sensitivities such that enzyme properties from the lower thermal variance site have a lower thermal denaturation temperature compared with high variance site - this points to higher thermal tolerance for sites with high thermal variance. (iii) The sediment temperature range selects for different taxa with different thermal properties (although this element of the study could be communicated more clearly - and from more in depth investigation).

This study brings together a large body of work - and at times from a reader perspective, it's difficult to follow. It touches on a number of important evolutionary principles of enzyme function and adaptation that are often studied in isolation. Thus, a strength - yet a co-occurring challenge - is to interpret this large, and complex volume of data.

I think that the work, following, perhaps additional work, provides a significant advance in environmental microbiology, and should help those interested in predicting climate change related sensitivities for ecosystems. By tapping into the natural esterase activities as an example, this work opens doors to understanding natural variations and thermal tolerances. The work extends the single enzyme/single organism approach used most often in thermal biology to a broader ecosystems perspective, which is valuable.

The manuscript tends towards overinterpretation of their findings. I agree the findings are important, but tempering the implications is also key. The title is bold; I am not convinced by the work here on a single enzyme (or enzyme class) and its thermal properties explain the plasticity of marine microbiomes. This is an oversimplification, in the least. Further just with respect to the title, the tense of "enzymes" should probably be singular. Perhaps the title could include ester hydrolase in the title?

I do not think that the abstract as a whole, accurately reflects the science performed. For example, nowhere does it state the enzyme class studied, or how these enzymes were actually studied. That enzymes have a temperature dependent response, in and of itself is not novel. Providing quantitative results would help.

The abstract concludes "temperature-driven enzyme selection is a planetary scale invariable mechanism that shapes microbiomes thermal plasticity".... And continues into a run-on sentence. However, I do not think that this study demonstrates this broad, global conclusion, that is based on results from studying esterase activities measured from marine sediment proteomes studied at three depths that experience different thermal variance ranges.

I have a few suggestions that may help clarify the manuscript, in particular through the precise use of terms. The conventions used in thermal tolerance, stenothermal vs. eurythermal have also been used for microbial enzymes and could be used to convey the variance/tolerance ranges. The concept of isozymes could also be explored with respect to the thermal variance aspects of the study. In addition, there are a number of terms used that are poorly defined, or ambiguous. There are many, but I will just mention a few that arise in the abstract. For example, in the abstract line 35 - "community-level enzyme adaptation" is vague and leaves reader wondering what that is; line 36 "enzyme thermal selection" is not clear - enzymes do not select. Line 37-38: "microbial enzyme temperature dependent profile" - should probably be temperature-dependent enzyme activity. Line 38: The concept of a "total active proteome" - conveys all the active enzymes, however this study has focused on hydrolytic esterase. Line 44: "cultural microbiomes" is ill-defined. In fact, the whole concept that microorganisms were cultured here is not an appropriate use of the term (methods/results and discussion). These issues persist through the manuscript.

There are also some points of clarification in terms of the science presented that should be addressed.

1. As reported in the conclusion line 333 "enzyme variants that are capable of being active under more variable temperatures" there are distinctions to clarify here with respect to the experiments conducted. The question comes to whether an individual esterase enzyme itself has a broad thermal tolerance or whether different isozymes (within the same, or different genomes) are active at different temperatures. This concept goes back in thermal studies led by Somero et al., Daniel et al., and others who did careful enzyme kinetic work. Further discussion of this is warranted.

2. Upon inspection Figs. 1B and C, I have a question about the differences observed in the MAT and max enzymatic activities. Can you explain the results in which there is not so much of a difference between MAT and max activity Temperature for the Irish Sea sample (~ 8 C), vs. a much larger difference (20-30 C) between samples from higher MAT sediments and their enzymatic activities? Similar relationship seen in Fig 1C. The relationship with 1D shifts - and slopes of these two lines are not the same. I also think that it's likely that there is a better curve to fit other than linear to the Tara dataset. Interpretation of that would also be helpful to understanding a more global thermal denaturation properties of the esterase enzymes.

3. I urge moderating claims from this study. For example, results from the thermal variance low/intermediate/high esterase activity profiles (e.g., Fig. 2e and f are not so different between the LTV and ITV, and ITV and HTV, yet only the differences in results are discussed). Likewise - see lines 324-326 still is referring to this data which studied esterases. How do other enzymes react? Comparisons to studies in the literature could benefit the discussion of the results.

4. The results and discussion points of the last section (lines 264-326) are difficult to sort through; recommend revising so that the results and message being conveyed are presented more clearly, and then are discussed with respect to the literature effectively.

5. Overall I think it would be useful for the authors to discuss what work needs to be done in this field to better understand the enzyme thermal sensitivities. It would be interesting to better address enzyme kinetics, sequence-based or structural differences in the polypeptides studied here as well. How variable are the enzymes measured in terms of their sequences? This is a useful body of work, with a lot of angles to study it from.

Specific points to consider for clarification.

Line 33: The abstract starts off with "multiorganismal communities" - the phrase is internally redundant.

Line 79: "condition" should be plural

Line 85: "cultivable bacteria" I recommend using a different term that more accurately reflects the experiment in which natural community was stimulated by growth in rich media; perhaps enriched heterotrophic bacterial fraction?

Line 91 and in conclusion: "temperature-driven enzyme selection" - I suggest clarification that conveys that microbial evolutionary processes select for enzymes.

Line 112: is the temperature being referred to here environmental temperature or experimental temperature?

Table 1. Clarify what bold represents; and why other models tested, in particular temperature and salinity, or temperature and pH in some cases, are not considered.

Fig. 1b-c: The data from lower temperature environments is sparse - it's not explained why the Td experiments were not tested with the Irish samples? I gather a metagenome was not prepared?

Line 158: this states the "same transect as above" however the Irish sample was not included. Please clarify.

Line 160: it would appear that the enzymatic activity data with model substrates data was not shown - would be useful to state that as then I understand that the Td work was preferred; but not clear how those two directly relate.

Line 177: what is the result referred to here in response to salinity?

Line 387 and elsewhere: Reiterate in the text the depths of the three sediment samples. 2, 25 and 50 meters. This also has ecological meaning that warrants interpretation.

Line 396: is citation to # 68 correct?

Lines 396-398: the methods here - though referencing other papers - could benefit from brief descriptions of the activities; in addition to providing some gauge for the reader to understand how many of the enzymes were discovered through the metagenomic library functional assays vs. in silico mining (that could also appear in the results)

Lines 403-404 and later - there are no details for gene synthesis; where was it performed etc.?

Line 515: suggest revising wording "metaphylogenomic" to taxonomic.

Overall - it would be useful to have all sequence data deposited to GenBank in one place.

Reviewer #3:

Remarks to the Author:

In the manuscript entitled "Enzymes adaptation to habitat thermal legacy explains the plasticity of marine microbiomes" the authors hypothesize that environmental temperatures influence marine microbiome plasticity through temperature-mediate enzyme selection, which can be indicators of thermal adaptation and/or "thermal legacy". The latter may reveal patterns of thermal adaptation that could aid in predictions of how a microbial community responds to temperature fluxuation. The study appears well designed and does present evidence that MAT and thermal variation influence the microbial community structure in sediment samples. However, there are some issues that should be addressed prior to publication.

- Line 97: Perhaps more info on the plasticity of ocean microbiomes would help orient readers. Plasticity in terms of adaptability or variability in constitution? Both?

- Line 106: why is ester-hydrolytic activity a "reliable proxy" of microbial physiology? Also, was only one enzyme utilized in this inquiry? Why not expand the panel to cross multiple functional categories? The bias mentioned in Line 124 could also be somewhat addressed by using a multi-enzyme panel to perform this temperature assessment.

- SDS-PAGE for metaproteome analysis is antiquated and not deep nor very descriptive. Counting spots is not sufficient for this analysis. Many spots can be the same protein but post-translationally modified, creating a new spot (or a series of spots) that shouldn't be double counted. Also, it's hard to make claims about the activity of proteins here (i.e. the active proteome). Dead, non-active cells will also have a measurable proteome. Is there a chance there are other protein sources in the sediment (i.e. plant matter/dead creatures) that would impact this 2D profile? These questions impact the assertion on Lines 127-128 that the gel "... revealed that they [sediments] were not dominated by a specific group/kind of proteins, but a wide range of them". I don't see how you can make this conclusion from a gel.

- Following the last comment, the LC-MS/MS metaproteome analysis is the better way to go here. However, the number of reported protein identifications are exceedingly low for complex microbial communities. Though proteome extraction in sediments can be tricky, I would expect a much greater depth of measurement (upwards of 5000-10000 proteins / 3000-5000 protein groups at least). These low numbers make it difficult to assess these data for protein distribution information and I don't see how this relative incompleteness addresses the mentioned bias nor makes me confident in the "Relative abundance" metric mentioned in Lines 132-137. Perhaps using only the metagenome information (i.e. number of hydrolases or abundance) is appropriate to normalize data to account for the bias mentioned in Line 124.

- Sup. Table 3: Were metagenomes and metaproteomes measured for all samples and locations along the MAT range? The T (°C) seems a little too consistent. If so, perhaps indicating the MAT for each locale would be beneficial, especially to assess whether or not temp was the main driver of hydrolase activity.

- Line 150: "... indicating the absence of a core-set of ester-hydrolyzing proteins among all the samples". This line seems to suggest that ester-hydrolytic activity may not be a "reliable proxy" as indicated earlier. I would still vote for a more expansive enzymatic panel to be studied. However, would adding a dendrogram/evolutionary analysis across all esterase sequences identified – and binning them by MAT – help substantiate your conclusions in Lines 151-153?

- Lines 250-252: Considering taxa profiles can help explain this plasticity (i.e. from an optimal growth temp perspective), what is driving what? Are the taxa that have more thermo-tolerant enzymes able to populate VT regions, or is the temperature driving an evolutionary shift in taxa/"novel" taxa? I think my issue is with the phrase "... by selecting more adapted enzymes". Isn't it less about the specific enzymes, and more about taxa (and ALL their enzymes) that are adapted to high temps and variability?

- Likewise, is there a differentiation between thermal structural integrity vs. a specific taxa's ability to maintain properly folded proteins? You do measure the structural stability, but only on one enzyme class. Are there other hallmarks of increased stability based on enzymatic sequence analysis like more cysteines for disulfide bridge formation? What about increases in a specific taxa's chaperone profile to maintain fold?

- Can you comment on the data in Sup Fig S9 (higher temps, lower richness/Alpha div) vs. what is presented in Fig 3C (taxonomic diversity)? On its surface, the data seem contradictory.

Subject: Author replies (in bold) to the Reviewers' comments on manuscript NCOMMS-21-51397.

RESPONSES TO REVIEWER #1 REMARKS TO THE AUTHOR

The study by Marasco et al explores the hypothesis that enzyme thermal selection explains the plasticity of marine microbiomes in response to temperature.

Understanding regulatory mechanisms of thermal stress response in the marine microbiome is a welcomed addition to the broad interests in the thermal plasticity and adaptability of microbiome facing risks of changing environmental conditions. Knowledge of the complexity of adaptations is crucial to infer which species will become winners or losers, and to predict resilience and tipping points. In addition, marine microbiomes have not been scrutinized to the same extent as their terrestrial counterparts so I encourage studies in this direction.

Below I suggest ways by which the authors could improve the interpretation and discussion of their results that might help the reader understand the significance of the data, as well as provide a landmark comparative framework for future studies. I think these minor edits would strength the manuscript but are not necessary for publication and can be done, if desired, at the authors discretion.

We thank the reviewer for finding value in our study and providing valuable comments and suggestions to improve the message of our manuscript. We have provided point-by-point responses to all comments and concerns. In the revised version of the manuscript (“Marked Up Manuscript”), the modifications in response to the comments of reviewer #1 have been highlighted in yellow.

My few minor comments are relative to some narrative aspects, putting in a bit more detail in the Introduction, and Results section.

INTRODUCTION: Perhaps add more nuance to the introduction of the differences between genetic adaptation and physiological plasticity – they are currently not introduced and should be explicitly entwined. Reading the Introduction, it feels like you're slightly underselling the importance that bacteria (short-generation time species) may also have high potential for timely genetic adaptation and produce new variants. The narrative should reconcile this complexity more clearly.

As suggested by the reviewer, we have reframed our introduction, including the differences between genetic adaptation and physiological plasticity, the complexity of bacterial

community genetic adaptation to changing conditions and the short bacterial generation time.

METHODS: Authors studied the thermal properties of 80 isolated enzymes from 14 sediments along the above mentioned latitudinal transect and 150 additional enzymes retrieved from 56 seawaters locations from Tara Ocean expedition. By quantifying the microbial enzyme temperature-dependent profiles in proteomes of marine sediments from the Irish Sea to the southern Red Sea authors recorded a significant temperature-dependent response of the enzymatic activity. The lack of general optimised methods of samples collection often produces results that may be controversial and do not indicate the true trend or introduce limits in data interpretation. It would have been more effective and simpler to draw conclusions if more homogenous samples would have been taken and analysed.

The reviewer points to a problem we all encounter when working with environmental samples: the analysis of homogeneous samples and their collection and processing using standardized methods. Although much work has been done to optimise and standardise sample collection protocols, this is an intrinsic bias over latitudes and locations dictated by the enormous environmental variability. This may also apply to the problem related to the extraction of DNA, RNA and proteins. Focusing on these important considerations, we agree with the referee that using homogeneous samples for comparative studies is necessary. For this reason, we have pointed our attention to marine sediments which present similar macroscopic characteristics over latitudes and locations and may be treated with uniform sampling and analytical procedures and protocols. We have further extended our investigation to the already available metagenomes from well-characterized and highly homogeneous seawaters (from 56 different locations) from the Tara Ocean Expedition. We synthesized and expressed enzymes from sequences retrieved from a metagenome database using standardized protocols and procedures on these sets of samples. We have discussed this important aspect in the revised manuscript.

This issue (bias) also refers to the cloning and activity of ester-hydrolases, some analysed in this study and some others retrieved from previous studies. I understand the difficulty in accessing samples at various latitudes in a FAIR way and I acknowledge the significance in publishing these results as they stand.

We thank the reviewer for this comment. There are two possible strategies to have a comprehensive enzyme dataset: on one end, there is the possibility of isolating, synthesizing and characterising proteins starting from metagenomic datasets of previous studies and on the other end is to include as many sites as possible. Since the latter approach, which we implemented on a latitudinal transect from the Irish Sea to the Southern Red Sea, was limited by our sampling capacity, to expand our range of exploration, we have added to the sediment set of samples from sites that we physically sampled, the sites from the Tara Ocean datasets, from which a set of sequences were retrieved and included in this study. All our

synthesized proteins (total, $n = 233$) were produced in soluble active form after expression to compare their activity consistently.

RESULTS DISCUSSION: The paper as written lacks clarity on the mechanisms of temperature compensation through natural selection (never mentioned) and physiological plasticity that are generally adopted for coping with changes of temperatures. Authors focus their conclusions exclusively on the changes of activity of enzymes as sign of thermal adaptation with no mention on changes in flexibility (the most common response to changing conditions at protein level). Changes in protein flexibility may play an important role at diverse temperatures without altering the global structure and the active site, thus reducing the selective pressure on new genes.

We thank the reviewer for this constructive comment, which allowed us to improve our manuscript. Indeed, we agree on the importance of flexibility constraints, and we have now performed structural and flexibility modelling and added the results of the flexibility analyses following the recently published protocol (Nutschel *et al.*, 2021 *J Chem Inf Model.* 61:2383-2395). Our results (new Figure 2e,f) show that the proteins in the dataset present rigidity/flexibility adapted to the thermal conditions. Overall, these findings indicate that esterases from organisms found in warmer environments are more rigid (less flexible) enzymes. These new results have been added to the revised manuscript.

Nature always produces cold-adapted proteins with decreased stability and viceversa. In addition, most studies indicate an inverse relationship between stability and activity while in your study it seems that there is a direct correlation between these two properties e.g. higher stability-higher activity.

We thank the referee for this comment, with which we partially agree. Indeed, we agree that microorganisms from cold environments generally contain proteins with decreased stability and low optimal temperature. In contrast, those from hot environments typically have proteins with increased stability and high optimal temperature. However, there are also examples that this relation is protein-dependent. For instance, the lipase from *Candida antarctica*, isolated from sediments at the bottom of the Antarctic Lake Vanda, perennially covered with 3–5 m of ice, is most active at 45°C and is able to work efficiently at > 90°C (Dominguez de Maria *et al.*, 2005 *J Molecular Catalysis B: Enzymatic* 37:36-46). Also, several enzymes from the Antarctic bacterium *Oleispira antarctica* isolated from the sea surface at 4.7°C are most active at 50°C; indeed, the fact that *O. antarctica* proteins exhibit generally higher temperature optima suggested a ‘warmer’ origin of this organism (Kube *et al.*, 2013 *Nature Comms* 4:2156). In our study, sites with non-extreme low temperatures were mostly included, so we cannot conclude much about the activity and stability of enzymes from cold environments. Having said that, we would like to point out that our study does not investigate the relationship between stability and activity but rather the relationships between the thermal regime of the environmental sites from which enzymes originated and their stability (measured as denaturing temperature and phase transition temperature) and optimal temperature for activity. The fact that this relationship has been found not only within

enzymes from different sites (new Figure 2) but also in the same site in different seasons (new Figure 3), and that it is connected to the higher or lower microbial growth at different temperatures (new Figure 4) reinforce our conclusions.

Some of the strategies developed at the protein level include (i) increasing of enzyme concentration (physiological plasticity), ii) temperature dependent expression of iso-enzymes (physiological plasticity, regulation of genes), and new variants of enzymes for which reaction rates tend to become temperature independent or adapted to new conditions (genetic adaptation, e.g. new genes). Authors have any functional evidence/information on the up- and/or down-regulated expression of genes encoding the enzymes selected in the study?

We thank the reviewer for this interesting comment. Through the analysis of 233 individual enzymes (228 esterases assigned to 9 different sub-families as reported in Supplementary Figure S7, and 5 extradiol dioxygenases) retrieved across a broad latitudinal gradient (from 62,2°S to 16°N) from at least 70 marine locations, and mean annual temperature (MAT) from -1.4°C to 29.5°C, our work suggests that microorganisms mainly adapt to different thermal conditions by the selection of thermally-adapted enzymes (through genetic adaptation, that is through new genes) rather than the selection of enzymes with broader thermal tolerances. From our data, the temperature-dependent expression of isozymes, e.g., to increase the concentration of enzymes for which reaction rates tend to become temperature-independent and thus work sub-optimally under a new condition (physiological plasticity, regulation of genes), could not be ruled out. The fact that the relationships herein described were found both in proteins extracted from environmental samples (where expression could play a role; Figure 1) and in individual enzymes from different sources (Figure 2) suggests that the level of expression may not be the main factor in the adaptation of microorganisms to different temperatures.

REFERENCES: The few references on protein thermal adaptation are old. More references on this should be included.

Thanks to the reviewer for the suggestion; additional references have been added to the revised manuscript.

RESPONSES TO REVIEWER #2 REMARKS TO THE AUTHOR

The research led by Marasco et al, studying the thermal tolerance of xylase hydrolytic activity across a wide variety of enzyme preparations from marine sediments and water columns differing in temperature was a large-scale effort aimed at uncovering underlying patterns in thermal adaptation and tolerance range. The work is novel through applying a broad biogeographic approach to enzyme thermal property characterizations, where it incorporates an ecological perspective that considers the thermal legacy of a site (and organisms therein) to interpret the thermal tolerance of the hydrolytic enzymes. Key results demonstrated that (i) enzymes/proteomes

exhibit a temperature dependent range of hydrolytic esterase activity that relates to their native habitat thermal properties, (ii) esterase activity from sites with different thermal variance ranges (low - intermediate - high) show different sensitivities such that enzyme properties from the lower thermal variance site have a lower thermal denaturation temperature compared with high variance site - this points to higher thermal tolerance for sites with high thermal variance. (iii) The sediment temperature range selects for different taxa with different thermal properties (although this element of the study could be communicated more clearly - and from more in depth investigation). This study brings together a large body of work - and at times from a reader perspective, it's difficult to follow. It touches on a number of important evolutionary principles of enzyme function and adaptation that are often studied in isolation. Thus, a strength - yet a co-occurring challenge - is to interpret this large, and complex volume of data. I think that the work, following, perhaps additional work, provides a significant advance in environmental microbiology, and should help those interested in predicting climate change related sensitivities for ecosystems. By tapping into the natural esterase activities as an example, this work opens doors to understanding natural variations and thermal tolerances. The work extends the single enzyme/single organism approach used most often in thermal biology to a broader ecosystems perspective, which is valuable.

We thank the reviewer for finding interest in our study and for the positive and supportive statements. We also appreciate the time spent to provide us with detailed comments and suggestions to improve the quality and clarity of the work. As proposed by the reviewer, we have performed additional work on different aspects of the study, including i) determination of enzyme flexibility and phase transition temperature, ii) determination of the temperature at which maximal enzymatic activities are found in the whole proteome extracted from the sediments, precisely that of six additional classes of enzymes other than esterases, iii) calculating the denaturation temperature, optimal temperature and phase transition temperature for the enzymes studied. We have provided point-by-point responses to all concerns, as reported below. Modifications to the text are highlighted in light blue in the revised version of the manuscript submitted as "Marked Up Manuscript".

The manuscript tends towards overinterpretation of their findings. I agree the findings are important, but tempering the implications is also key.

We thank the reviewer for the comment. We revised the entire manuscript to avoid overinterpretation of our results.

The title is bold; I am not convinced by the work here on a single enzyme (or enzyme class) and its thermal properties explain the plasticity of marine microbiomes. This is an oversimplification, in the least. Further just with respect to the title, the tense of "enzymes" should probably be singular. Perhaps the title could include ester hydrolase in the title?

We thank the reviewer for the constructive suggestion that prompted us to extend our investigation to other enzyme classes to test if the observed findings are related to a single enzyme class or can be extended to other enzymes. We have performed further analyses on six additional enzymatic classes: phosphatases, beta-galactosidases, nucleases,

transaminases, aldo-keto reductases and extradiol dioxygenases. The relationship between mean annual temperature (MAT) and the enzyme activity found on esterases was also confirmed for the other six enzymatic classes with the same trends. These results consolidate our conclusions that enzyme thermal properties explain the thermal plasticity of marine microbiomes. We propose retaining the original manuscript title with these additional data.

I do not think that the abstract as a whole, accurately reflects the science performed. For example, nowhere does it state the enzyme class studied, or how these enzymes were actually studied. That enzymes have a temperature dependent response, in and of itself is not novel. Providing quantitative results would help.

As suggested, we have improved the abstract to describe the work done and results obtained more synthetically and comprehensively.

The abstract concludes “temperature-driven enzyme selection is a planetary scale invariable mechanism that shapes microbiomes thermal plasticity”.... And continues into a run-on sentence. However, I do not think that this study demonstrates this broad, global conclusion, that is based on results from studying esterase activities measured from marine sediment proteomes studied at three depths that experience different thermal variance ranges.

We concur with the reviewer that the analysis of esterase activities may not be sufficient to provide an overview of the global microbial metabolism. We selected esterases as a target because of their abundance and presence across microorganisms (at least one per genome), wide environmental distribution, and important physiological functions. However, as requested by the reviewer and the editor, we have performed further work and generated novel data. In the revised version of the manuscript, we have also provided the results obtained from the analysis of six other enzyme classes, extending the breadth of the findings. They all showed significant correlations with MAT of the sites of origin, analogously to esterases. This has been added to the revised version of the manuscript (Figure 1). In addition, we would like to point out that among the cloned marine enzymes obtained from the Irish Sea–Red Sea transect, we have also included a set of extradiol dioxygenases (EXDO) from the Mediterranean that are structurally and catalytically different to esterases. Optimal temperature and denaturing temperature of EXDO are now reported as Supplementary Figure S4 ($n = 5$; T_d : $R^2 = 0.97$, $p = 0.0025$; T_{opt} : $R^2 = 0.85$, $p = 0.27$).

In consideration of the comment of the reviewer, we have now toned down the claims in relation to the planetary-scale conclusions.

I have a few suggestions that may help clarify the manuscript, in particular through the precise use of terms. The conventions used in thermal tolerance, stenothermal vs. eurythermal have also been used for microbial enzymes and could be used to convey the variance/tolerance ranges. The concept of isozymes could also be explored with respect to the thermal variance aspects of the study.

We thank the reviewer for the suggestion about the terms stenothermal and eurythermal. We thought carefully about the suggestion, however, we do not feel comfortable using the two terms because, in our datasets, we cannot sort out how many enzymes or microorganisms respond to the thermal regimes within a community.

Isoenzymes (or isozymes) are enzymes that differ in amino acid sequence (from point mutations to large variations) and thus usually have different kinetic parameters but catalyze the same chemical reaction. Commonly, isozymes are variants of the same enzyme that result from the existence of more than one gene locus in the same genome. So, technically speaking, we provided here information about: (1) different enzymes as we provided information about esterases, extradiol dioxygenases, phosphatases, transaminases, beta-galactosidases, nucleases and aldo-keto reductases; and (2) different enzymes variants as we provided information about esterases (or ester hydrolases) that differ in their sequences. Given the low pairwise sequence similarity among all sequences (16.7% for the set of sequences of the Irish Sea, the Mediterranean and the Red Sea, and 38.1% for the Tara Ocean expedition dataset), we prefer to use the term ‘enzyme variants’ rather than ‘isozymes’.

In addition, there are a number of terms used that are poorly defined, or ambiguous. There are many, but I will just mention a few that arise in the abstract. For example, in the abstract line 35 - “community-level enzyme adaptation” is vague and leaves reader wondering what that is; line 36 “enzyme thermal selection” is not clear - enzymes do not select. Line 37-38: “microbial enzyme temperature dependent profile” - should probably be temperature-dependent enzyme activity. Line 38: The concept of a “total active proteome” - conveys all the active enzymes, however this study has focused on hydrolytic esterase. Line 44: “cultural microbiomes” is ill-defined. In fact, the whole concept that microorganisms were cultured here is not an appropriate use of the term (methods/results and discussion). These issues persist through the manuscript.

We thank the reviewer for the comment. We revised the manuscript to clarify the terms used according to the specific suggestions. We have implemented the changes in the abstract and the whole manuscript.

There are also some points of clarification in terms of the science presented that should be addressed.

1. As reported in the conclusion line 333 “enzyme variants that are capable of being active under more variable temperatures” there are distinctions to clarify here with respect to the experiments conducted. The question comes to whether an individual esterase enzyme itself has a broad thermal tolerance or whether different isozymes (within the same, or different genomes) are active at different temperatures. This concept goes back in thermal studies led by Somero et al., Daniel et al., and others who did careful enzyme kinetic work. Further discussion of this is warranted.

We thank the reviewer for this comment, and we acknowledge the fundamental thermal studies on enzyme kinetics of the authors mentioned. We consider that our data add to those pioneering studies the fact that the same relationship of esterase activity with temperature

was found both for pools of proteins extracted from environmental samples where different esterase variants occur (Figure 1) and for individual enzyme variants from different sources (Figure 2). This observation suggests that the main factor in the adaptation to the temperature of the enzymes (and the microorganisms that harbour them) is the selection of the enzyme variants with thermal characteristics adapted to the specific site conditions. Such selection of enzyme variants adapted to the site temperature may derive from mutations and the formation of new proteins or the enrichment of variants that arise as soon as the temperature conditions allow, for instance, and among others, through organism physiological adjustment or species turnover. From our data, we cannot identify which of these mechanisms prevail, but we conclude that the selection of adapted enzyme variants contributes to drive the functional plasticity of the microbial communities. We have discussed these aspects in the revised version of the manuscript.

2. Upon inspection Figs. 1B and C, I have a question about the differences observed in the MAT and max enzymatic activities. Can you explain the results in which there is not so much of a difference between MAT and max activity Temperature for the Irish Sea sample (~ 8 C), vs. a much larger difference (20-30 C) between samples from higher MAT sediments and their enzymatic activities? Similar relationship seen in Fig 1C. The relationship with 1D shifts - and slopes of these two lines are not the same. I also think that it's likely that there is a better curve to fit other than linear to the Tara dataset. Interpretation of that would also be helpful to understanding a more global thermal denaturation properties of the esterase enzymes.

We thank the reviewer for this interesting observation. We do not have a general answer to the observed trend on esterases because after performing additional experiments on the other six independent enzyme classes (please, see the answers to the editor and to the specific questions of reviewers #2 and #3 on the opportunity to test new enzyme classes) we observed that the difference between MAT and the max activity temperature varies according to the enzyme class (new Figure 1). While changes in such difference occurred for esterases, phosphatases, beta-galactosidases and aldo-keto reductases, this was not the case for transaminases, extradiol dioxygenases and nucleases. We conclude that this is a protein-dependent feature.

Regarding the regression to fit the Tara Ocean dataset, we thank the reviewer for raising this point. To improve the analysis of the dataset and study the non-linear relationship between enzymatic responses and temperature, we performed a piecewise regression and calculated the breakpoint (flexus) where the slope of the regression significantly changed. We observed a breakpoint at MAT of 25.1°C-27.7°C , which identifies the differences pointed out by the reviewer. The result shows that below the temperature breakpoint, the esterases have a more similar response (indicated by the low slope of the line) to temperature increase. In contrast, over the MAT breakpoint, esterases present a steeper response (indicated by the steeper slope) to temperature increase. This offers the advantage of increasing species' adaptation to higher temperatures, where the effects on cell viability are stronger compared to lower temperatures. This has now been mentioned in the revised manuscript.

3. I urge moderating claims from this study. For example, results from the thermal variance low/intermediate/high esterase activity profiles (e.g., Fig. 2e and f are not so different between the LTV and ITV, and ITV and HTV, yet only the differences in results are discussed). Likewise - see lines 324-326 still is referring to this data which studied esterases. How do other enzymes react? Comparisons to studies in the literature could benefit the discussion of the results.

The data shown in the Figure 3d,e (new version) refer to the activities of esterases extracted from HTV, ITV and LTV sediments, while the panels f and g refer to the thermal unfolding patterns of proteins (stability) measured by circular dichroism and expressed as degrees of ellipticity (θ). Even though we have shown that the general thermal response of esterases (Figure 1) is mirrored in six other enzyme types, we agree with the reviewer and have moderated the claims by referring to esterases that were the enzymes studied in these specific experiments. Also, according to the suggestion, we have further revised our manuscript by considering the studies in the literature.

4. The results and discussion points of the last section (lines 264-326) are difficult to sort through; recommend revising so that the results and message being conveyed are presented more clearly, and then are discussed with respect to the literature effectively.

We revised the results and discussion as suggested.

5. Overall I think it would be useful for the authors to discuss what work needs to be done in this field to better understand the enzyme thermal sensitivities. It would be interesting to better address enzyme kinetics, sequence-based or structural differences in the polypeptides studied here as well. How variable are the enzymes measured in terms of their sequences? This is a useful body of work, with a lot of angles to study it from.

We thank the reviewer for this suggestion. We have revised the discussion by highlighting additional aspects to deepen the understanding of the enzyme thermal response.

Specific points to consider for clarification.

Line 33: The abstract starts off with “multiorganismal communities” - the phrase is internally redundant.

We modified it to “organismal community”.

Line 79: “condition” should be plural

We modified the text as suggested.

Line 85: “cultivable bacteria” I recommend using a different term that more accurately reflects the experiment in which natural community was stimulated by growth in rich media; perhaps enriched heterotrophic bacterial fraction?

We acknowledge the reviewer’s suggestion to use a more precise definition. We have adopted the term “enriched heterotrophic bacteria” and modified the text accordingly.

Line 91 and in conclusion: “temperature-driven enzyme selection” - I suggest clarification that conveys that microbial evolutionary processes select for enzymes.

We have followed the suggestion and pointed to the microbial evolutionary processes that select the enzymes.

Line 112: is the temperature being referred to here environmental temperature or experimental temperature?

It referred to the mean annual temperature (MAT) of the site; this has now been specified.

Table 1. Clarify what bold represents; and why other models tested, in particular temperature and salinity, or temperature and pH in some cases, are not considered.

We thank the reviewer for the opportunity to clarify this point. Bold represents the lower Akaike Information Criterion (AIC) obtained by running different models and indicates the best model we obtained by testing the different factors, namely temperature, salinity and pH. These results showed that, in all three datasets, the best model that describes the enzymatic response is the one including temperature (MAT) and no other factors, justifying our focus only on temperature throughout the paper.

Fig. 1b-c: The data from lower temperature environments is sparse - it's not explained why the Td experiments were not tested with the Irish samples? I gather a metagenome was not prepared?

We understand the observation about the scarcity of data on the Irish Sea samples. This was because no metagenome was prepared when the original version of this paper was submitted. We have now provided sequence information of a metagenome from the Menai Bridge site at the Irish Sea and we have successfully identified, synthesized, and characterized a new set of five esterases, whose features were added and discussed. We specify that the synthesized enzymes were not obtained from the original Irish Sea sediment sample (MAT 12°C) but from a 12-months enrichment culture at 20°C. Since we have found that the MAT influences the selection of enzymes, we then considered as MAT for these enzymes 20°C instead of 12°C. Therefore, the MAT for the Irish Sea, Mediterranean Sea, and Red Sea sites ranged from 18.30°C to 24.75°C. However, we would like to point out that the esterases from the Tara Ocean expedition were retrieved across a broad latitudinal gradient (from 62,2°S to 16°N) with MATs from -1.4°C to 29.3°C, thus covering a wide range of temperatures including low-temperature environments (see Supplementary Table S7).

Line 158: this states the “same transect as above” however the Irish sample was not included. Please clarify.

In the original version of the manuscript, we have not included samples from the Irish Sea because we had a metagenome only from a microcosm maintained at 20°C for one year, so not entirely representative of the natural condition of the Menai Strait in the Irish Sea. However, given the availability of such metagenome, we have now included the

characterization of esterases (optimal temperature, denaturing temperature and phase transition temperature) from the microcosm, considering a MAT value of 20°C in the data analysis. The new version of the manuscript now includes for the Irish sample: i) the evaluation of the activity of seven enzyme classes (see new Figure 1) in the total active protein extracted from the Menai Strait sediments (MAT, 12°C); ii) a set of five additional esterases identified within metagenome sequence from a microcosm prepared with the sediments of the Menai Strait incubated for one year at 20°C (new Figure 2). Starting from these sequences, we performed gene synthesis, expression, and characterization of these five esterases. For the data analyses of these esterases, we have considered a MAT of 20°C. We have clarified this point in the revised version of the manuscript, and we have stated in the caption of the new Figure 2 that the Irish sea esterases were obtained from a microcosm incubated for one year at 20°C and not directly from the metagenome of the collected sediments.

Line 160: it would appear that the enzymatic activity data with model substrates data was not shown - would be useful to state that as then I understand that the Td work was preferred; but not clear how those two directly relate.

We thank the reviewer for the comment. In the revised version of the manuscript, we have now included the enzymatic activity of total proteins extracted from each sample, with model substrates (new Figure 1 shows the T_{opt} and Supplementary Table S2 details the raw data where the specific activity at each temperature has been reported). In the case of single enzymes, we have now reported the T_{opt} (panels a and c); the specific activity of each enzyme at pH 8 and T_{opt} are given in Supplementary Table S6 (column O) and Supplementary Table S7 (column Q).

Line 177: what is the result referred to here in response to salinity?

We did not find a statistically significant relationship between the esterase properties and salinity, but only with temperature. We have revised the text of the paragraph.

Line 387 and elsewhere: Reiterate in the text the depths of the three sediment samples. 2, 25 and 50 meters. This also has ecological meaning that warrants interpretation.

As per the suggestion, we have reported in the results and methods the depths of the three sediment samples. We have also commented on the ecological meaning of the environmental differences among the three sites, and we pointed out that ecologically major parameters such as hydrostatic pressure, sunlight and MAT have no or negligible differential effects on the sediments, differently from TV.

Line 396: is citation to # 68 correct?

We thank the reviewer for spotting this mistake. We have now included the correct citation.

Lines 396-398: the methods here - though referencing other papers - could benefit from brief descriptions of the activities; in addition to providing some gauge for the reader to understand how many of the enzymes were discovered through the metagenomic library functional assays vs. *in silico* mining (that could also appear in the results).

We acknowledge the reviewer's suggestion, and we have added this information in the method section, where a description of the origin of each of the enzymes herein reported was summarized. In brief, from the total set of enzymes herein reported, 28 were retrieved by functional screening of the corresponding pCCFOS metagenomics libraries and 205 (55 from the Irish Sea, Mediterranean and the Red Sea; 150 from Tara Ocean Expedition) by *in silico* mining (homology sequence analysis) of the meta-sequences generated after sequencing of the DNA material from resident microbial communities in each of the samples.

We have also included a scheme that summarizes the synergetic workflow used in the present study, from the recovery of protein extracts and their analysis to the enzyme selection, cloning and expression (see Supplementary Method S1). In addition to the above, we have included in the results section and Supplementary Material (see Supplementary Result S1 and Supplementary Figure S7) a description of the different families and sub-families to which each of the enzymes reported is assigned. Briefly, the 78 esterases from the Irish Sea, Mediterranean and the Red Sea included sequences with a typical α/β hydrolase fold and conserved G-X-S-X-G motif (Family [F]I, 6; FIV, 35; FV, 18; FVI, 3; FVII, 2; FX, 2), one sequence belongs to true lipase sub-family, one sequence with a serine beta-lactamase-like modular (non- α/β hydrolase fold) architecture and a conserved S-X-X-K motif (FVIII), and an additional set of nine sequences were assigned to the meta-cleavage product (MCP) hydrolase family, with typical α/β hydrolase fold. All 150 ester-hydrolases from Tara Ocean belong to FIV. This diversity and the low pairwise sequence similarity among all sequences (16.7% for the set of sequences of the Irish Sea, the Mediterranean and the Red Sea, and 38.1% for the Tara Ocean set) suggests that the diversity of polypeptides herein investigated is not dominated by a particular type of protein or highly similar protein clusters but consists of diverse nonredundant sequences assigned to multiple folds, families and subfamilies, which are distantly related to known homologues in many cases.

Lines 403-404 and later - there are no details for gene synthesis; where was it performed etc.? **Thanks for pointing out this missing information. This has now been added.**

Line 515: suggest revising wording "metaphylogenomic" to taxonomic.
Done.

Overall - it would be useful to have all sequence data deposited to GenBank in one place. **Done.**

RESPONSES TO REVIEWER #3 REMARKS TO THE AUTHOR

In the manuscript entitled “Enzymes adaptation to habitat thermal legacy explains the plasticity of marine microbiomes” the authors hypothesize that environmental temperatures influence marine microbiome plasticity through temperature-mediated enzyme selection, which can be indicators of thermal adaptation and/or “thermal legacy”. The latter may reveal patterns of thermal adaptation that could aid in predictions of how a microbial community responds to temperature fluctuation. The study appears well designed and does present evidence that MAT and thermal variation influence the microbial community structure in sediment samples. However, there are some issues that should be addressed prior to publication.

We thank the reviewer for carefully reading our manuscript and providing valuable feedback. We have provided point-by-point responses to all concerns, as reported below. Modifications to the text are highlighted in green in the revised version of the manuscript submitted as “Marked Up Manuscript”.

Line 97: Perhaps more info on the plasticity of ocean microbiomes would help orient readers. Plasticity in terms of adaptability or variability in constitution? Both?

We acknowledge the need for clarity in the manuscript highlighted by the reviewer for better framing of our study for the reader. We intend plasticity as the extent of variability of the community response to temperature resulting from multiple mechanisms, including adaptability (physiological adaptation or genetic adaptation) and variability in the constitution (species turnover). All such mechanisms may occur at the same time. We have included this explanation and supported it with literature references.

Line 106: why is ester-hydrolytic activity a “reliable proxy” of microbial physiology? Also, was only one enzyme utilized in this inquiry? Why not expand the panel to cross multiple functional categories? The bias mentioned in Line 124 could also be somewhat addressed by using a multi-enzyme panel to perform this temperature assessment.

With new work and analyses, we now show that very similar trends of response to temperature hold for seven diverse functionally independent enzymatic classes (esterases, extradiol dioxygenase, phosphatase, beta-galactosidase, nuclease, transaminase and aldo-keto reductase, new Figure 1), showing that esterases are a reliable proxy of microbial enzyme response to temperature. We thank the reviewer for the comment that prompted us to extend the study to other enzymes to substantiate the representativeness of esterases in studying the enzyme response to temperature. The new data reinforce the concept of the previous rationale that considered the following features of esterases to be used as a proxy of thermal studies of enzymes: i) esterases are essential for microbial cell functioning and present in all microbial genomes, ii) esterases present a high abundance with at least one per genome and are widely distributed in the environment, iii) esterases play important physiological functions, iv) consolidated procedures are available for their identification, expression and characterization. We believe that the extension of the measures to the other six enzymatic classes gives a more robust overview of microbial metabolism and makes our

conclusions more solid. Based on the new data in Figure 1, we have used only esterases for the following experiments with single purified enzymes from the latitudinal transect and the Tara Ocean expedition dataset.

SDS-PAGE for metaproteome analysis is antiquated and not deep nor very descriptive. Counting spots is not sufficient for this analysis. Many spots can be the same protein but post-translationally modified, creating a new spot (or a series of spots) that shouldn't be double counted. Also, it's hard to make claims about the activity of proteins here (i.e. the active proteome). Dead, non-active cells will also have a measurable proteome. Is there a chance there are other protein sources in the sediment (i.e. plant matter/dead creatures) that would impact this 2D profile? These questions impact the assertion on Lines 127-128 that the gel "... revealed that they [sediments] were not dominated by a specific group/kind of proteins, but a wide range of them". I don't see how you can make this conclusion from a gel.

We thank the reviewer for this comment which points out a critical issue. About the enzymatic diversity, and to ascertain whether the results could be influenced by the protein extraction method and the bias in the number of proteins extracted and of the esterases responsible for the activity in the samples, we have performed meta-proteomic and metagenomic analyses. Results have demonstrated that the differences in enzyme activity of the active proteomes may have a biological significance and is not due to a bias in the extraction protocol applied. We also agree with the reviewer that SDS-PAGE and spot counting for metaproteome analysis are antiquated, not deep, nor very descriptive. Therefore, we performed in the original submission shot-gun proteomic analysis as a complement.

According to the reviewer's observation, on which we agree, we have modified the sentence to inform the reader that, based on shot-gun proteomic analysis and not only on 2D gel profiles, the sediments are not dominated by specific groups of proteins.

Following the last comment, the LC-MS/MS metaproteome analysis is the better way to go here. However, the number of reported protein identifications are exceedingly low for complex microbial communities. Though proteome extraction in sediments can be tricky, I would expect a much greater depth of measurement (upwards of 5000-10000 proteins / 3000-5000 protein groups at least). These low numbers make it difficult to assess these data for protein distribution information and I don't see how this relative incompleteness addresses the mentioned bias nor makes me confident in the "Relative abundance" metric mentioned in Lines 132-137. Perhaps using only the metagenome information (i.e. number of hydrolases or abundance) is appropriate to normalize data to account for the bias mentioned in Line 124.

Recent studies investigating the metaproteome of marine sediments using different protocols (Wöhlbrand *et al.*, 2017 *Proteomics* doi:10.1002/pmic.201700241) were able to identify up to 250 proteins per gram of sediment sample from the Pacific Ocean and the North Sea, a number below the ones herein found. The treatment-specific efficiency of protein extraction was attributed to the different chemical properties of the applied detergents. However, since

in this study we aimed to obtain functional protein, we avoided using detergent to lyse the microbial cells during the extraction protocol.

Regarding normalization, we have reported in Supplementary Table S4 not only the absolute numbers of proteins, but also their relative abundances (%) referred to the total number of proteins in the corresponding metagenomes; by doing this, we have normalized the data to account for the bias in the protein extraction and metagenome sequencing effort.

Sup. Table 3: Were metagenomes and metaproteomes measured for all samples and locations along the MAT range? The T (°C) seems a little too consistent. If so, perhaps indicating the MAT for each locale would be beneficial, especially to assess whether or not temp was the main driver of hydrolase activity.

We agree, and MAT has been included in the revised Supplementary Tables S6 and S7 and in the related Figure 2.

Line 150: "... indicating the absence of a core-set of ester-hydrolyzing proteins among all the samples". This line seems to suggest that ester-hydrolytic activity may not be a "reliable proxy" as indicated earlier. I would still vote for a more expansive enzymatic panel to be studied. However, would adding a dendrogram/evolutionary analysis across all esterase sequences identified – and binning them by MAT – help substantiate your conclusions in Lines 151-153? **Thanks for this comment which highlights that this sentence was not clear. Through the analysis of the metagenomes, we found a total of 843 sequences potentially encoding esterases, none present in all samples and only 73 shared among a few samples (Supplementary Figure S4). Their relative abundance, compared to the total ORFs, ranged from 0.10% to 0.29% (average: 0.22%; interquartile range [IQR]: 0.08%; Supplementary Table S3), with no linear correlation with the MAT of the sampling site ($p = 0.098$; $R^2 = 0.39$). This tells us that the relative abundance of esterase is relatively similar in all the sites and what changes are the esterase response to temperature.**

As indicated in our answers to the previous comments, to confirm that esterases are a reliable proxy for studying the thermal response of microbial community enzymes, we have generated the data for six other activities, namely extradiol dioxygenase, phosphatase, beta-galactosidase, nuclease, transaminase and aldo-keto reductase, which are now included in the revised manuscript as Figure 1.

According to the suggestion, we have built a phylogenetic tree including all esterase sequences (with the indication of the families to which each affiliated) identified and tested ($n = 227$; 78 from the Irish sea, the Mediterranean and the Red Sea transect and 150 from Tara Ocean datasets; Supplementary Table S6 and S7) and binning them by MAT (see new Supplementary Figure S7). Through this analysis, we concluded that the large sequence divergence does not identify any clustering pattern associated with MAT.

Lines 250-252: Considering taxa profiles can help explain this plasticity (i.e. from an optimal growth temp perspective), what is driving what? Are the taxa that have more thermo-tolerant

enzymes able to populate VT regions, or is the temperature driving an evolutionary shift in taxa/"novel" taxa? I think my issue is with the phrase "... by selecting more adapted enzymes". Isn't it less about the specific enzymes, and more about taxa (and ALL their enzymes) that are adapted to high temps and variability?

We thank the reviewer for this comment, which points out two considerations. On one side, taxa that have enzymes with wider thermal tolerance will be able to populate sites with a larger TV. On the other side, those microorganisms that contain thermally adapted enzymes can have more opportunities to be enriched in sites with a larger TV. We consider that asking what comes first is a sort of circular question that, unfortunately, is difficult to answer. The only interpretation of our data that we are comfortable adopting is that whatever of the two processes come first and is prevailing (yet coexisting), the net result of the thermal selection should favour more thermally suited enzymes. Considering the two processes of i) taxa substitution and ii) selection of more thermally resistant variants, our data suggest that they coexist, but the acclimation of the microbial community to TV prevails over a community compositional change (Figure 4).

Likewise, is there a differentiation between thermal structural integrity vs. a specific taxa's ability to maintain properly folded proteins? You do measure the structural stability, but only on one enzyme class. Are there other hallmarks of increased stability based on enzymatic sequence analysis like more cysteines for disulfide bridge formation? What about increases in a specific taxa's chaperone profile to maintain fold?

We thank the reviewer for these comments. The analysis of disulfide bridges and taxa chaperone profiles is interesting, and the real occurrence of those properties would deserve experimental tests with appropriate inhibitors (disulfide bridges). It may not reflect enhanced thermal properties, while the taxonomic assignments of chaperons at genus and species levels cannot be unambiguously achieved. For these reasons, we preferred to focus on the analysis of the protein structural rigidity that has also been suggested by reviewer #1. In the revised version of the manuscript, we have included such additional work (new Figure 2).

Can you comment on the data in Sup Fig S9 (higher temps, lower richness/Alpha div) vs. what is presented in Fig 3C (taxonomic diversity)? On its surface, the data seem contradictory.

Supplementary Figure S9 refers to the temperature decay relationships of similarities (Bray-Curtis) and alphadiversity (richness and evenness) obtained by analysing the total bacterial communities in the sediments from 16S rRNA gene amplicon sequencing; while Figure 4 (Figure 3 in the first version of the manuscript) report data referred to bacteria extracted from the sediments and cultivated in the laboratory at different temperature (*i.e.*, the "enriched heterotrophic bacterial fraction"). The results illustrated in Supplementary Figure S9 tell us that the diversity of the whole sediment bacterial community does not strongly change according to the sites and their increasing temperatures (82% of the OTUs are thermal generalists found under all the three TVs) and that there is a relatively limited

decline of diversity (around 10% of alphadiversity). Figure 4 (old Figure 3) is referred to a subfraction of the whole bacterial community, that is, the heterotrophs from the site sediments that are enriched in microcosms in a rich medium at different temperatures and show that no substantial changes in the taxonomic types of bacteria occur, but differences were observed in term of growth, *e.g.*, *Bacillus* members were consistently enriched in all the three sediments at 40°C, however, only those from HTV sediments grew up to high OD levels. The two datasets indicate that the LTV, ITV and HTV sediments host similar microbial communities, but such microorganisms have differentially adapted enzymes that support their growth at different temperature conditions.

Reviewers' Comments:

Reviewer #1:

Remarks to the Author:

The authors seem to have answered most of my queries satisfactorily.

Reviewer #2:

Remarks to the Author:

The substantial modifications in the revised manuscript have improved the manuscript significantly. The results are more clearly communicated, and with the additional research conducted, the work has a larger impact. This is a multifaceted, complicated study which addresses important concepts in thermal tolerance, ecosystem variability and adaptation. I think the methods figure - Method S1 will be of great help to the readership as the reader will be able to cross-reference the figure when going through different results, and the different sources of the enzymes.

The authors did a nice job of addressing reviewer comments, in making a number changes to address clarifications, explain the science, and conduct additional analyses.

In particular, the addition of analyses to include six additional enzyme classes helps convey the significance of the trends observed. The results shown in revised Figure 1 are compelling. In that similar trends were observed across a suite of diverse, ubiquitous enzyme families. Likewise, I found the details provided in explaining the significance of the response to temperature being dictated by MAT helpful, and think that the science is more clearly conveyed (lines 258-262).

The thermal variability results more clearly describe the relationship between thermal legacy and thermal plasticity, while the investigation into species composition was also better addressed.

I do not have any major suggestions for modification, and overall am satisfied with the response to reviewers.

I have a few remaining minor comments: (note that the manuscript will benefit from careful English grammar editing beyond the few cases mentioned below)

Title - I think that the term Enzymes' could just as easily be "Enzyme" in the title and convey the same meaning.

Line 40: Organismal community is still an odd term as here the manuscript really addresses microbial communities, so I'd suggest focusing the term being used.

Line 46-57: Unclear to me if the reader will understand what recorded a significant mean annual temperature (MAT)-dependent response is - a few words of clarification might help

Line 57: I think that the worked microbiome should be singular

Line 58: process should be "processes"

Line 127-129: English usage could be improved by flipping the sentence around some so that it say "we quantitatively evaluated (i) and (ii) in sediments with different MAT.

Line 246: although the point here is that the relationships found for protein rigidity the relationship for the Tara Oceans data set is not significant for the data prior to the inflection; and is a fairly poor relationship thereafter (albeit apparently significant - with $R^2=0.1$) I would suggest further

interpreting the relationship at temperatures less than 20 degrees C - is there a reason that Tp would not vary at lower temperatures?

Lines 548-552, Supp. Table S7. Are the AA sequences for the TARA Oceans data set those that were identified in the data sets; or the sequence following codon refactoring for E. coli expression? If not - I'm not quite sure what is preferred by the journal, but it might be important to report the synthesized sequences. Also - it would be helpful to note the manufacturer source of the synthetic DNA sequences.

Reviewer #3:

Remarks to the Author:

Upon re-review of the revised manuscript, I am ready to recommend publication in Nature Communications. The authors have adequately addressed all of my comments. I am particularly impressed by their expanded enzymatic profile that extended and strengthened their argument that MAT and TV influences a microbiome's enzymatic thermal plasticity.

Subject: Author replies (in bold) to the Reviewers' comments on manuscript NCOMMS-21-51397A.

RESPONSES TO REVIEWER #2 REMARKS TO THE AUTHOR

The substantial modifications in the revised manuscript have improved the manuscript significantly. The results are more clearly communicated, and with the additional research conducted, the work has a larger impact. This is a multifaceted, complicated study which addresses important concepts in thermal tolerance, ecosystem variability and adaptation. I think the methods figure - Method S1 will be of great help to the readership as the reader will be able to cross-reference the figure when going through different results, and the different sources of the enzymes. The authors did a nice job of addressing reviewer comments, in making a number changes to address clarifications, explain the science, and conduct additional analyses.

In particular, the addition of analyses to include six additional enzyme classes helps convey the significance of the trends observed. The results shown in revised Figure 1 are compelling. In that similar trends were observed across a suite of diverse, ubiquitous enzyme families. Likewise, I found the details provided in explaining the significance of the response to temperature being dictated by MAT helpful, and think that the science is more clearly conveyed (lines 258-262). The thermal variability results more clearly describe the relationship between thermal legacy and thermal plasticity, while the investigation into species composition was also better addressed.

I do not have any major suggestions for modification, and overall am satisfied with the response to reviewers.

We thank reviewer #2 for carefully reading the revised version of our manuscript and providing valuable feedback. We have provided point-by-point responses to all concerns, as reported below and we have included in the revised manuscript corrections according to the suggestions of reviewer #2. Modifications to the text are highlighted in yellow in the revised version of the manuscript submitted as a "Marked Up Manuscript".

I have a few remaining minor comments: (note that the manuscript will benefit from careful English grammar editing beyond the few cases mentioned below).

To improve the reading and grammar of our revised manuscript, we have submitted it to a Scientific Editor at the Research Publication Services of the King Abdullah University of Science and Technology. Please, see below the certificate of language editing.

EDITORIAL CERTIFICATE

Date: 18 December 2022

Manuscript Authors: R Marasco, M Fusi, C Coscolín, A Barozzi, D Almendral, R Bargiela, C Gohlke, C Pflieger, J Dittrich, H Gohlke, R Matesanz, S Sanchez-Carrillo, F Mapelli, T Chernikova, P Golyshin, M Ferrer, D Daffonchio

Title: Enzyme adaptation to habitat thermal legacy explains the plasticity of marine microbiomes

To whom it may concern:

This letter confirms that the manuscript corresponding to the information detailed above was edited by a professional, native English-speaking editor at KAUST.

We guarantee language accuracy as delivered to the author(s) on the above date. We have not altered the intent or research content as written by the author(s). The author(s) may accept or reject any of our comments or suggestions upon receipt of the document edited.

Sincerely,

Jenny Booth

Scientific Editor
Research Publication Services
King Abdullah University of Science and Technology

Title - I think that the term Enzymes' could just as easily be "Enzyme" in the title and convey the same meaning.

We have modified the title as suggested.

Line 40: Organismal community is still an odd term as here the manuscript really addresses microbial communities, so I'd suggest focusing the term being used.

As suggested by the reviewer, we have modified the sentence and used “microbial community” from the beginning.

Line 46-57: Unclear to me if the reader will understand what recorded a significant mean annual temperature (MAT)-dependent response is - a few words of clarification might help.

As suggested, we have revised the sentence and clarified this point.

Line 57: I think that the worked microbiome should be singular.

Done.

Line 58: process should be “processes”

Done.

Line 127-129: English usage could be improved by flipping the sentence around some so that it say “we quantitatively evaluated (i) and (ii) in sediments with different MAT.

Done.

Line 246: although the point here is that the relationships found for protein rigidity the relationship for the Tara Oceans data set is not significant for the data prior to the inflection; and is a fairly poor relationship thereafter (albeit apparently significant - with $R^2=0.1$) I would suggest further interpreting the relationship at temperatures less than 20 degrees C - is there a reason that T_p would not vary at lower temperatures?

We thank the reviewer for the comment, and we have now discussed the possible reasons why the regression before $T \sim 20^\circ\text{C}$ in Figure 2f was not significant. We consider that such lack of regression before $T \sim 20^\circ\text{C}$ can be interpreted with evolutionary trade-offs observed during biochemical adaptation to lower temperatures where enzymes have to keep a minimum phase transition for correct functioning, while those adapted to higher temperatures have the possibility to increase it to cope with the higher metabolic requirements of the organisms (Hochachka and Somero, 2002. Biochemical adaptation mechanism and process in physiological evolution, Oxford Univ. Press, New York). However, we have also to consider that the protein rigidity and phase transition temperature (T_p) has been determined by the CNA method based on constraint/interaction networks in the protein. This analysis is applied to conformational ensembles generated by MD simulations at room temperature, possibly leading to an evening out of T_p values for esterases that transition around this temperature, *i.e.*, systems with a T_p at or below room temperature might all be influenced similarly by loosening their bonding network. By contrast, systems with a transition temperature at or above room temperature would still be discriminated against. This support the fact that we saw a significant correlation in Figure 2e, where the

lowest MAT is ~19°C, but not when esterases from environments with MAT below this temperature were analysed, such as in Figure 2f for the Tara Ocean dataset. We would like to underline that this is not a shortcoming of the MD simulation/CNA combination but rather reflects its sensitivity.

Based on these considerations, we have clarified this point and discussed and interpreted the data, considering biological and methodological points of view.

Lines 548-552, Supp. Table S7. Are the AA sequences for the TARA Oceans data set those that were identified in the data sets; or the sequence following codon refactoring for *E. coli* expression? If not - I'm not quite sure what is preferred by the journal, but it might be important to report the synthesized sequences. Also - it would be helpful to note the manufacturer source of the synthetic DNA sequences.

We thank the reviewer for this comment that allows us a more precise description of the procedures used in the experiments. As suggested, we have added in Supplementary Tables S6 and S7 the AA sequences of the synthesized sequences once introduced in the expression vector. We have also added in the Materials and Method the manufacturer source and slightly modified the text as follows: “Once identified, the sequences encoding the wild-type enzymes here identified and reported for the first time from all the geographically distinct locations (including the ones from the Tara Ocean Expedition) were used as templates for gene synthesis. Genes were codon-optimized to maximize expression in *E. coli*. Genes were flanked by BamHI and HindIII (stop codon) restriction sites and inserted in a pET-45b(+) expression vector with an ampicillin selection marker (GenScript Biotech, EG Rijswijk, Netherlands). This plasmid, which was introduced into *E. coli* BL21(DE3), supports the expression of N-terminal histidine (His) fusion proteins, with the final amino acid sequences of all synthetic proteins being MAHHHHHHVGTGSNDDDDKSPDP-X (where X corresponds to the original sequence of the target enzyme, as detailed in Supplementary Tables S6 and S7)”.